# Unravelling the interplay of nitrogen nutrition and the *Botrytis cinerea* pectin lyase BcPNL1 in modulating *Arabidopsis thaliana* susceptibility

Antoine Davière[1,2,4], Aline Voxeur [1,4] ✉, Sylvie Jolivet[1], Luka Lelas[1,2], Samantha Vernhettes [1],
Marie-Christine Soulié[1,3,4] & Mathilde Fagard [1,4]

In this study, we investigated the interplay between nitrogen nutrition and the pectin degradation dynamics during Arabidopsis and Botrytis interaction. Our findings revealed that infected detached leaves from nitrogen-sufficient plants released more pectin lyase (PNL)-derived oligogalacturonides compared to nitrogen-deficient ones. We then focused on BcPNL1, the most highly expressed Botrytis *PNL* upon infection. Using mutant strains lacking *BcPNL1*, we observed reduced pathogenicity, a delay in germination and a lag in triggering the plant defense response. Additionally, in nitrogen-sufficient detached leaves, the elevated expression of jasmonic acid repressor genes observed upon infection with the wild-type strain was abolished with the mutants. These results linked the increased production of BcPNL-derived products to the increased expression of jasmonic acid repressor genes, contributing partially to the higher susceptibility of nitrogen-sufficient detached leaves. These findings could lay the foundation for new strategies aimed at reconciling both crop resistance to pathogens and the improvement of nitrogen nutrition.

The ascomycete *Botrytis cinerea* is a widely spread necrotrophic fungus responsible for the grey mold disease in over 1000 plant species[1]. As a necrotrophic pathogen, its infectious process relies on several synergistic molecular mechanisms, involving the secretion of a vast array of molecules that include cell wall-degrading enzymes, toxins, reactive oxygen species, peptidases, and acidifying compounds, which have been reported to contribute to its pathogenicity[2].

The availability of the sequenced genome of this fungus[3], has greatly facilitated genetic studies and the identification of putative pathogenicity factors. However, among the virulence factors, some are encoded by multigene families such as Carbohydrate-Active enZYmes (CaZY), which are subject to functional redundancy and compensation phenomena that make it challenging to pinpoint the roles of individual *B. cinerea* genes in pathogenicity. Consequently, only a few genes have been described as having a major role in pathogenicity[2]. Among them, some transcription factors and kinases have been identified, such as sucrose non-fermenting protein kinase 1 (BcSNF1) and the pH regulator BcPACC[4,5]. Furthermore, chitin synthases like Bcchs3a are also important pathogenicity factors,

influencing fungal adhesion to the host and plant defense mechanisms[6]. In addition to these, several secreted factors, such as the botrydial and botcinic acid toxins whose production involve the *BcBOT* and *BcBOA* gene clusters, as well as cell death-inducing proteins like BcXIG1 and BcCRH1, play significant roles in pathogenicity[7–9]. However, recent papers have raised concerns about studying pathogenicity factors exclusively in optimum conditions since plant-pathogen interactions are strongly impacted by abiotic factors[10,11]. Indeed for *B. cinerea*, we were able in a previous work to identify undescribed pathogenicity factors by varying plant nitrogen supply which significantly affect the interaction[12].

During infection, *B. cinerea* degrades the cell wall of the plants it infects. To achieve this, the pathogen releases numerous enzymes, with the most abundant belonging to the pectinase family[13]. Pectinases enable the pathogen to penetrate and spread within host tissues by degrading pectins, one of the major components of the plant cell wall barrier. The primary content of pectin, homogalacturonan (HGA), consists of α-1,4-linked residues of D-galacturonic acid (GalA), which can be methyl (Me)- and/or acetyl-esterified (Ac). Four types of HGA with different Degrees of

[1]Université Paris-Saclay, INRAE, AgroParisTech, Institut Jean-Pierre Bourgin (IJPB), Versailles, France. [2]Ecole Doctorale 567 Sciences du Végétal, Univ Paris-Sud, Univ Paris-Saclay, Orsay, France. [3]Sorbonne Universités, UFR 927, Paris, France. [4]These authors contributed equally: Antoine Davière, Aline Voxeur, Marie-Christine Soulié, Mathilde Fagard. ✉e-mail: aline.voxeur@inrae.fr

Methylesterfication (DM) can be distinguished: the fully methylesterified (DM close to 100%), the highly methylesterified (DM ~ 70%), the low methylesterified (DM ~ 30%) and the fully demethylesterified homo-galacturonans also called polygalacturonic acids. The degradation of these different HGA by pectinases leads to the production of oligogalacturonides (OG) with varying degrees of polymerization (DP) and methyl-acetylesterification. Among pectinases, pectin lyases (PNLs, CAZy family PL1), of which five are predicted in *B. cinerea*, degrade fully methylesterified pectins in which leaf epidermal cells are enriched[14]. BcPNL1 is the most expressed pectinase during the prepenetration and penetration stages on hard, wax-coated surface[15] and is likely the primary OG-producing pecti-nases of *B. cinerea* during early infection events fin tomato fruit[16] and *Arabidopsis thaliana* leaves[17] (Supplementary Fig. 1A). PNLs employ β-elimination resulting in highly methyl- and acetylesterified OGs of DP > 4 harboring an unsaturated bond at the non-reducing end[18] (Supplementary Fig. 1C). Concomitantly, fungal esterases, including pectin methylesterases (PMEs, 3 predicted in *B. cinerea*, CAZy family CE8) deesterify both the large methylesterified oligosaccharides and highly methylesterified pectins. Some fungi can also deacetylate pectins using pectin acetylesterases (PAEs) but no PAE is predicted in the *B. cinerea* genome[19,20]. Next, depolymerases degrade partially and fully methylesterified pectins and further break down the large PNL-derived oligosaccharides into smaller ones. These pectinases are categorized into two groups: hydrolases with polygalacturonases (Endo-PGs and Exo-PGs, 6 and 3 predicted respectively, CAZy family GH28) which can cleave demethylesterified sites of HGA backbone releasing both low or un- methylesterified OGs (Supplementary Fig. 1C) and lyases with pectate lyases which cleave fully demethylesterified pectins (Endo-PLs and Exo-PLs, 2 predicted, CAZy family PL1) releasing unsaturated non methylesterified OGs. It is worth to note that BcPG1, which is the most expressed PG upon growth of the lesion beyond the inoculation spot[21], where pectins are less methylesterified[14], and BcPG2 are crucial patho-genicity factors[22,23] (Supplementary Fig. 1B). PGs collectively represent 20% of the total protein content in the early secretome of *B. cinerea*[13].

Plants perceive the oligogalacturonides (OGs) produced by these pectinases as signals of damaged or modified self, enabling them to act as Damage-Associated Molecular Patterns (DAMPs) in Pattern-Triggered Immunity (PTI)[24]. This signaling pathway triggers several modifications in plant cells, leading to the accumulation of defense proteins and the production of hormones that enhance the defense reaction. Recently, Davidsson et al. (2017)[25] showed that short OGs (DP3) can efficiently trigger hormonal immunity mediated by jasmonic acid (JA) and salicylic acid (SA). These short OGs resulting from PG activity effectively protected *A. thaliana* from *Pectobacterium carotovorum* infection and are detected in *A. thaliana* leaves infected by *B. cinerea*[17]. Interestingly, research has also shown that OG oxidases (OGOX) secreted by the plant itself can inactivate OG-induced defense, thus preventing excessive plant cell death and infection during *B. cinerea* infection[26]. The hydrogen peroxide ($H_2O_2$) produced during OG oxidation mediates the activation of other defense processes, such as lig-nification or the inactivation of auxins, through the action of class III plant peroxidases[27]. Finally, early reports on OG activity demonstrated that long-chain OGs as a result of PG activity on fully demethylesterified HGA (DP10-15) serve as defense elicitors and can induce phytoalexin production[28,29]. However, we failed to detect these long-chain OGs in *A. thaliana* leaves infected by *B. cinerea*[17] as well as in infected tomato fruit[16]. Instead, we detected unsaturated methyl- and acetylesterified OGs with long DP capable of suppressing JA-mediated defense activation by up-regulating the pool of transcripts encoding negative regulators of JA signaling[17]. However, the subsequent degradation of these OGs by PMEs and PGs leads notably to the production of unsaturated methyl and acetylesterified tetragalacturonides ($GalA_4MeAc-H_2O$), which, in the end, strongly elicits JA-related plant defense. It is worth to note that, to date, neither the impact of OG methy-lesterification and DP on their activity is fully clear yet, nor is it known whether the presence of unsaturated bonds affects OG activity.

To gain deeper insight into the strategies employed by *B. cinerea* to successfully infect plants, we investigated the role of BcPNLs in

pathogenicity within the context of environmental modulation of the plant-pathogen interaction, specifically through nitrogen fertilization. Our research provides compelling evidence that nitrogen fertilization influences pectin degradation in *A. thaliana* detached leaves infected by *B. cinerea*. Through mutant and complementation analyses, we demonstrate the cru-cial role of BcPNL1 in increasing the susceptibility of plants grown with high nitrogen nutrition by inducing the expression of repressors of JA signaling. However, BcPNL1 alone cannot fully explain the connection between nitrogen fertilization and disease susceptibility. Our data confirm the involvement of BcPNL1 in producing unsaturated methyl-acetylesterified OGs with long DP, which are capable of suppressing the JA-mediated defense activation. In contrast, the *ΔBcpnl1* mutant shows a delay in ger-mination and in the establishment of plant defense response. In high nitrogen nutrition at a late phase of infection with the mutant, there is still a down-regulation of JA-repressor genes which are correlated with the absence of unsaturated methyl-acetylesterified OGs with long DPs. Alto-gether, our findings reinforce the fundamental role of BcPNL1 in mod-ulating plant susceptibility under high N conditions.

## Results

### Host nitrogen nutrition impacts pectin degradability during A. thaliana/B. cinerea interaction

First, we assessed if the host nitrogen nutrition impacts the OG turnover (i.e OG production and subsequent degradation into monomers) during infection by analyzing the OG produced in presence of the WT strain B05.10 on detached leaves from plants grown either with 0.5 mM (low N) or 10 mM $NO_3^-$ (high N) which correspond physiologically to nitrogen-deficient or -sufficient regimes respectively[12]. We decided to conduct all experiments on living detached leaves. Although this biological model does not reflect actual infection conditions as the disease development has been shown to be slower than on attached leaves[30], it offers the possibility to incubate plant leaves in a minimal liquid medium containing fungal spores that facilitates the col-lection of large amounts of OGs, allowing their structural characterization by a high-resolution mass spectrometry (HRMS)[17]. To do so, we incubated plant leaves in a liquid medium containing fungal spores of the WT strain B05.10, and subsequently analyzed the cell wall breakdown products released into the medium using HRMS[17]. It is worth noting that the minimal liquid medium used contains 25 mM $NO_3^-$. Consequently, the phenomena we observed correspond to the impact of N nutrition on the plant, notably its cell wall structure, that in turn affects the infection process rather than the impact of N availability on fungal virulence. Since only a few OGs have been shown to be detected at 12 h post-inoculation (hpi), we chose to perform this experiment a bit later, at 15 hpi. Interestingly, we found about 10% more PNL products at high N compared to low N with notably a higher quantity of PNL products of DP6, 7, and 9 at high N compared to low N by about 45%, 26% and 85%, respectively. This resulted in an increased proportion of PNL products and a decreased proportion of PG products, particularly for DP3, 5, and 6 (Fig. 1A). The discrimination between OGs resulting from PNL or PG activity is possible thanks to the specific β-elimination-mediated cleavage of HGA employed by PNLs (Supplementary Fig. 1C).

We next assessed if host nitrogen nutrition could impact fungal pec-tinase gene expression. We monitored fungal pectinase gene expression at 1-day post-inoculation (dpi), a time point when *BcPG1*-encoding gene expression has already been shown to be impacted by plant host nitrogen nutrition[12]. As depicted in Supplementary Fig. 2, *BcPG1,2* and *BcPNL1* stand out as the most highly expressed genes within the *BcPG* and *BcPNL* gene families, respectively, at both high and low N. *BcPNL2, 3, 4* were more than 10-fold less expressed compared to *BcPNL1* and *BcPNL5* was not expressed at all. Moreover, we observed that at 1 dpi on detached leaves and at 15 hpi in liquid culture with detached leaves, the expressions of *BcPNL1* were respectively decreased by 1.5 at high N in infected detached leaves and not significantly different (Supplementary Fig. 2A, B) in liquid culture although we detected more PNL products (Fig. 1A). As a consequence, the variations in PG and PNL products cannot be fully explained by transcriptional reg-ulation of the *PG* and *PNL* encoding genes. We then hypothesized that this

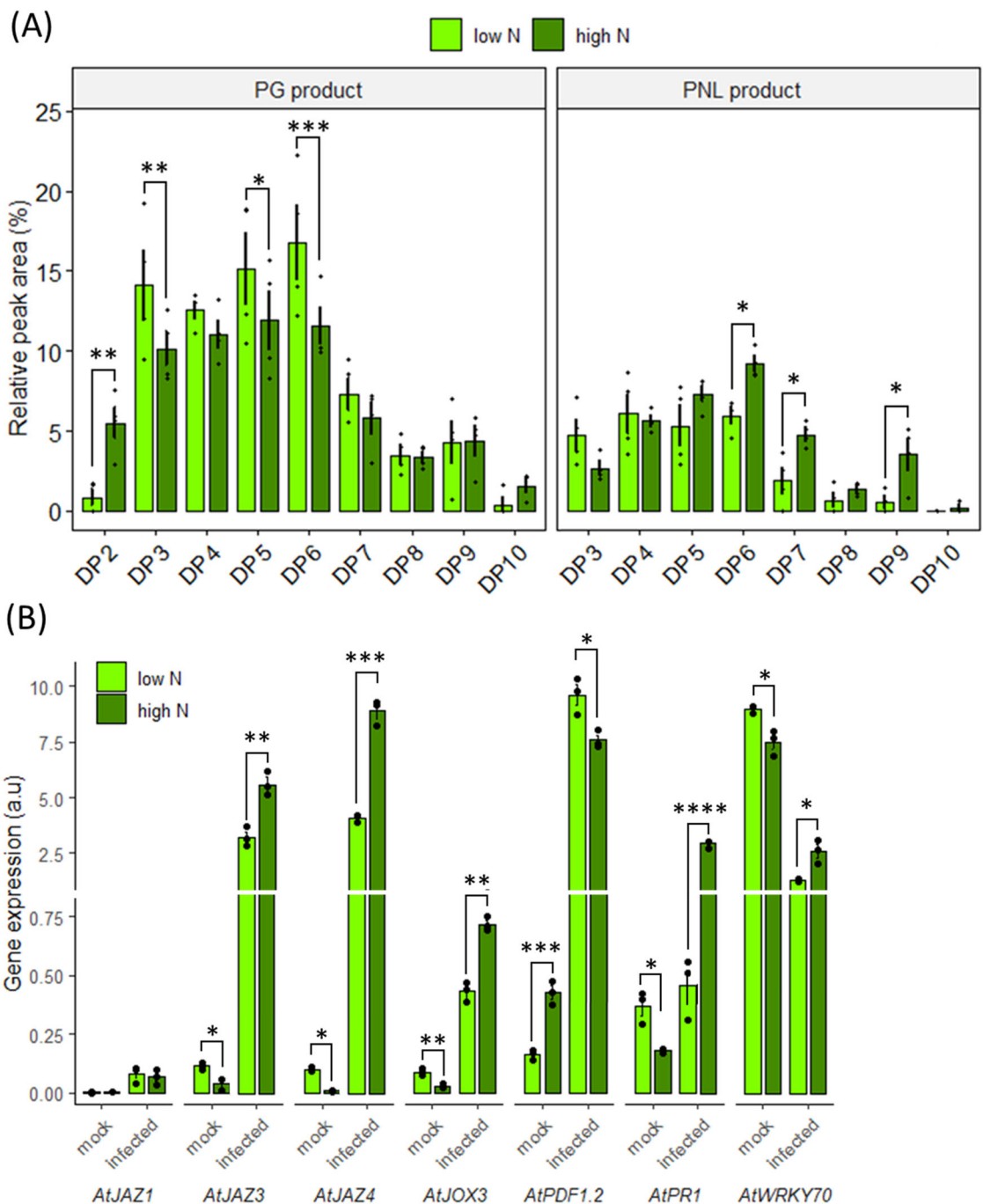

**Fig. 1 | Impact of host nitrate nutrition on *Botrytis cinerea* pectinolytic capacities and plant defense regulation. A** LC–MS quantitative analysis of OG production after incubation of spores from the WT strain of the fungus for 15 h with *A. thaliana* Col-0 leaves from plants cultivated at high or low N. The graph represents the peak area of each type of product over the total of all detected OGs expressed as %. **B** Transcript accumulation during infection in low N and high N conditions at 1-day post-inoculation on detached leaves are represented for plant defense genes. Plant gene targets were normalized with *AtUBI4* and similar results were obtained using *AtAPT1*. Three independent experiments were conducted with similar results. Data are expressed as mean normalized expression in arbitrary units (a.u.) and are the means of triplicates (±SE). Statistical differences represented between high and low N are the results of an ANOVA followed by a Fisher's LSD test in (**A**) and a two-sample t-test in (**B**): *$p < 0.05$, **$p < 0.01$, ***$p < 0.001$, ****$p < 0.0001$.

increased PNL activity in high N conditions could be caused by differences in pectin chemistry. One important feature for pectin degradation is its esterification status, so we tested whether PNL activity could be affected by methyl- and acetylesterification levels. Interestingly, we could only detect PNL products when the fungus was grown *in* vitro with methyl-acetylesterified pectins and not with methylesterified pectins suggesting that the presence of acetylesterification is a key factor modulating PNL activity (Supplementary Fig. 3A, B). Then, in a time course experiment on detached leaves from 12 hpi to 18 hpi, we observed that methyl-acetylesterified PNL products and methylesterified PG products were respectively increased and reduced in high N conditions compared to low N whereas unesterified and acetylesterified products were similar (Supplementary Fig. 3C). Altogether, these results suggest that the differences in PNLs mediated degradation upon fungal infection between high and low N detached leaves are due to differences in the organisation and/or abundance of methyl-acetylesterified regions in HGA.

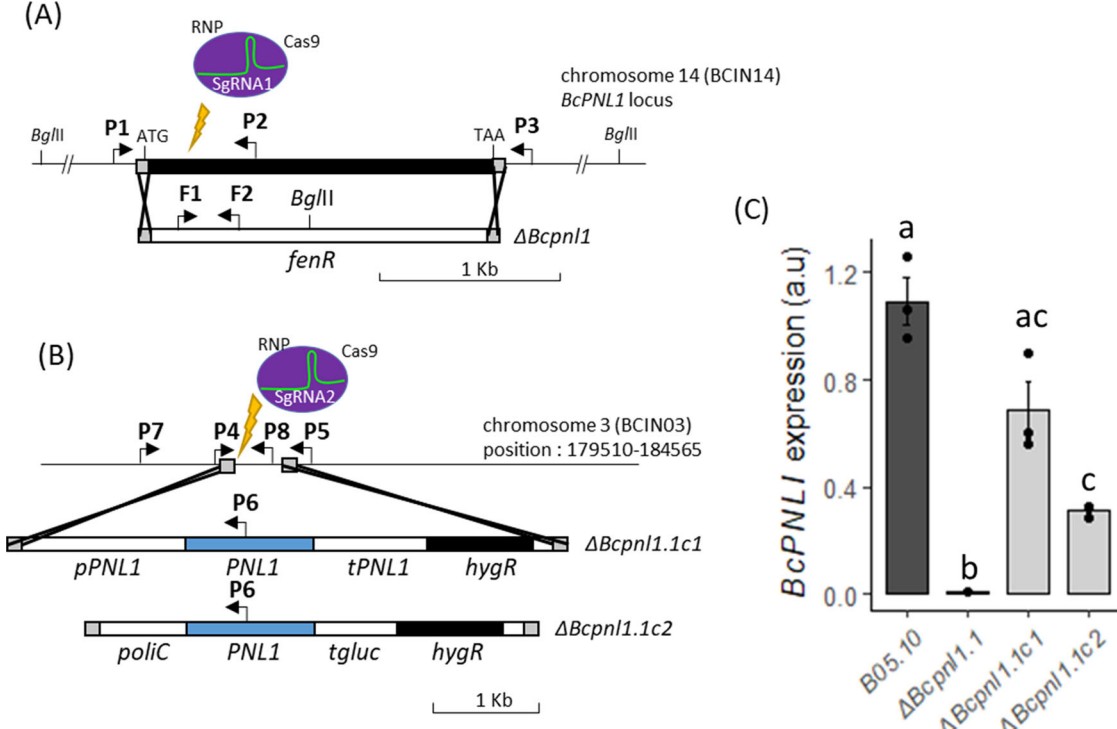

**Fig. 2 | Targeted gene replacement of *BcPNL1* in *Botrytis cinerea*.** Representation of the deletion strategy used to obtain *Bcpnl1* mutants (**A**) and complemented strains (**B**). Arrows indicate primers P1–P6 used for diagnostic PCR and primers F1 and F2 used to obtain the *fenR* probe to confirm absence of ectopic integration by Southern blot (Supplementary Fig. 4). **C** Transcript accumulation of the *BcPNL1* gene at 1 dpi on *A. thaliana* Col-0 leaves (high N) with the different strains of *B. cinerea*. *BcActA* was used for normalization of *BcPNL1* expression. Data are expressed as mean normalized expression in arbitrary units (a.u.) and are the means of triplicates (±SE) Letters represent the results of a two-sample t-test, $p < 0.05$.

Since PNL products of DP > 4 have been shown to up-regulate *JOX* and *JAZ* genes[17] respectively involved in repression of the JA pathway by inactivation of JA through hydroxylation[31,32] and repression of JA-induced genes[33], we searched for differences in nitrogen-dependent *JAZ* and *JOX* regulation during infection. Since it has been shown that the majority of changes in plant gene expression occurs by 24 hpi when the pathogen has penetrated the leaf epidermis, and that, naturally, OG production precedes transcriptomic activation[34], we chose to delay the transcriptomic analysis slightly compared to the OG analysis and performed it at 1 dpi. We found a significant higher expression of *AtJAZ3,4* and *AtJOX3* in high N compared to low N at 1 dpi with increases of 1.7-, 2.0-, and 1.7-fold, respectively (Fig. 1B). This was consistent with the higher production of high DP PNL products upon high N detached leaves infection (Fig. 1B). Accordingly, the JA marker gene *AtPDF1.2* was less expressed in high N conditions which was also previously observed in Soulie et al.[12]. Since SA and JA defense signalling pathways are mutually antagonistic, we also searched for SA marker genes[35]. Upon infection, we observed a 6-fold increase in *AtPR1* expression at high N and a 2-fold increase in the transcription factor *AtWRKY70* which was shown to be activated by SA and repressed by JA although its regulation is more complex[36,37].

### Targeted gene deletion of the BcPNL1 gene in B. cinerea leads to reduced pathogenicity at high N and low N

Next, to draw a clear connection between the higher accumulation of PNL products and the downregulation of the JA defense signaling pathway in high N compared to low N detached leaves, we investigated the function of *BcPNL1* which stands out as the most highly expressed gene within the *BcPNL* gene family as depicted in Supplementary Fig. 2. To investigate its function, we replaced the *BcPNL1* coding region with a deletion cassette containing the fenhexamid resistance gene, which was amplified from pTEL-fenh[38]. We employed PEG/CaCl$_2$-mediated transformation of *B. cinerea* protoplasts, in conjunction with the recently described CRISPR-Cas9 system, utilizing the ribonucleoprotein complex direct delivery strategy (Fig. 2A)[38]. *In* vitro, we selected fenhexamid-resistant transformants.

The absence of the wild-type *BcPNL1* gene and the correct integration of the cassette in each isolate were determined through PCR and confirmed via Southern blot using a DIG-labeled probe. Supplementary Fig. 4A demonstrates that out of 20 resistant transformants, we successfully identified three homokaryotic mutant strains, designated Δ*Bcpnl1.1*, Δ*Bcpnl1.2*, and Δ*Bcpnl1.3*. Others, such as the Δ*Bcpnl1.4* transformant, were found to be heterokaryotic. Nevertheless, all tested strains were devoid of ectopic integration, confirming the precision of the CRISPR-Cas9 system (as shown in Supplementary Fig. 4B). Subsequently, we selected the mutant strains Δ*Bcpnl1.1*, Δ*Bcpnl1.2*, and Δ*Bcpnl1.3* for further phenotypic characterization.

We next complemented the Δ*Bcpnl1.1* mutant with the *BcPNL1* gene either under the control of the native promoter (1.5 kb upstream of the ORF) or the strong *oliC* promoter from *Aspergillus nidulans*, resulting in the Δ*Bcpnl1.1c1* and the Δ*Bcpnl1.1c2* strains respectively (Fig. 2B and Supplementary Fig. 4C). The complementation cassettes were integrated at the same intergenic region of chromosome 3 using CRISPR-Cas9[39]. RT-qPCR analysis of the complemented lines revealed that the level of *BcPNL1* expression was only partially restored compared to the B05.10 strain (Fig. 2C).

First, we investigated whether the deletion of *BcPNL1* would affect the growth of the fungus in vitro when cultivated on solid medium in the presence of different types of substrates. Although we observed slight differences between the mutant strains, we observed no significant impact of *BcPNL1* deletion either on radial growth with Czapeck minimal medium supplemented with glucose or citrus highly methylesterified pectin as the sole carbon sources or on in vitro spore production on PDA (Supplementary Fig. 5).

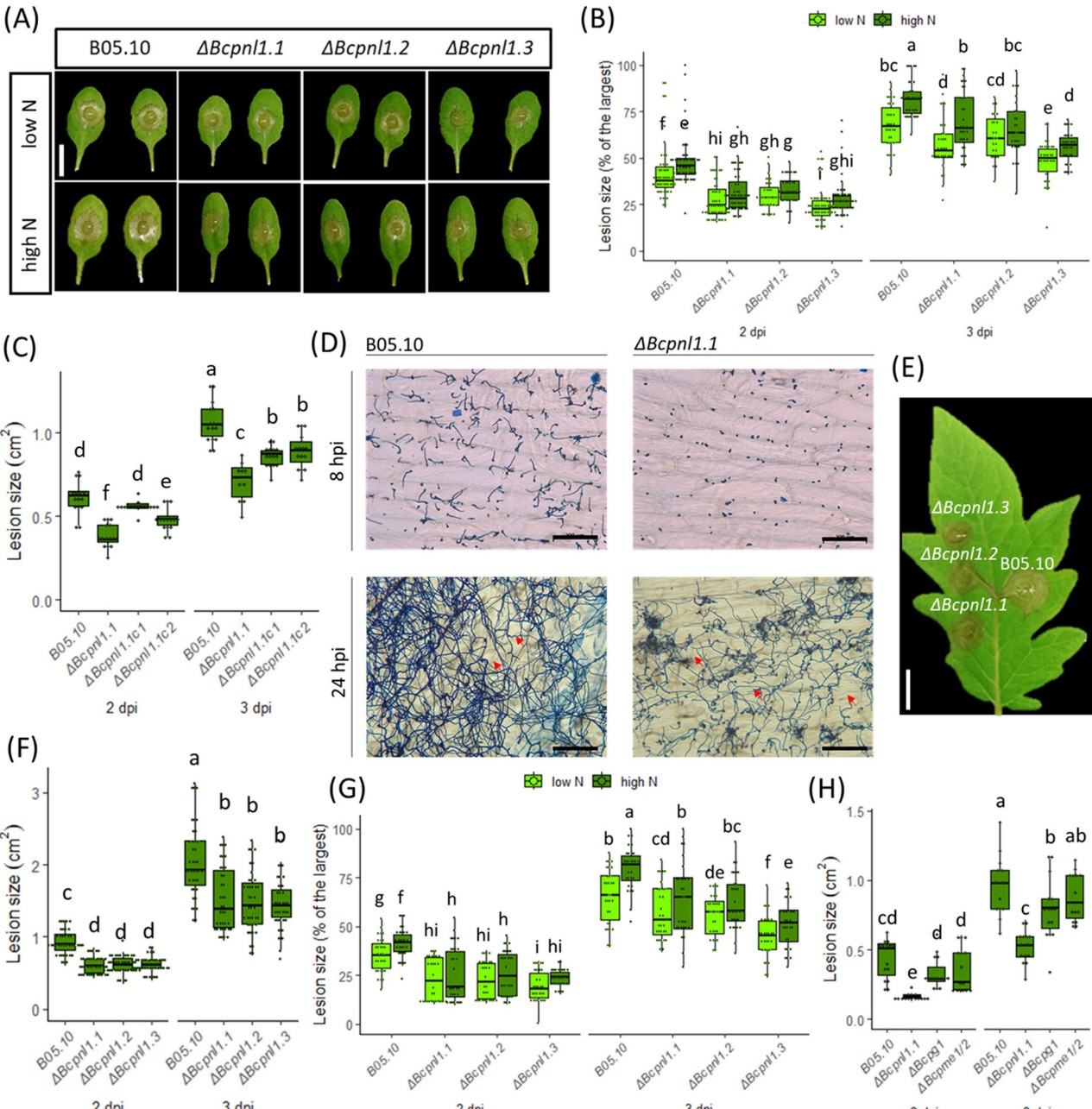

**Fig. 3 | The *ΔBcpnl1* mutant strains showed a strongly reduced pathogenicity on detached leaves that is partially restored in the complemented strains. A** Col-0 *A. thaliana* detached leaves at 2dpi. Scale bar = 1 cm. **B** Plot of the lesions at 2 and 3 dpi on Col-0 *A. thaliana* detached leaves. Plants were either grown with high or low nitrate supply (low N, high N). **C** Plot of the lesions on *A. thaliana* detached leaves at 2 and 3dpi in high N with B05.10, the *ΔBcpnl1.1* mutant and the two complemented strains *ΔBcpnl1.1c1* and *ΔBcpnl1.1c2*. **D** Coton blue coloration of the fungus on onion epidermis. Onion epidermises were inoculated with spore droplets of either B05.10 or the *ΔBcpnl1* mutant strains in ½ PDB and incubated for 8 h (top panel) or 24 h (lower panel) before coloration and observation under the microscope. Red arrows show penetrating hyphae. Since results were the same with *ΔBcpnl1.1/2/3*, only the observations for *ΔBcpnl1.1* are presented. Scale bar = 200 μm. **E** *S. lycopersicum* detached leaves, 2 dpi. **F** Plot of the lesions on tomato detached leaves at 2 and 3 dpi. Tomato plants were grown in high N (10 mM $NO_3^-$). Scale bar = 1 cm. **G** Plot of the lesions at 2 and 3 dpi on Wassilewskija *A. thaliana* detached leaves. **H** Plot of the lesions at 2 and 3 dpi on Col-0 *A. thaliana* detached leaves in high N condition with B05.10, the *ΔBcpnl1.1* mutant and two other pectinase mutants of the fungus *ΔBcpg1* and *ΔBcpme1/2*. Letters represent the results of an ANOVA test followed by a Fisher's LSD for multiple comparison ($p < 0.05$). Lesion sizes are expressed in cm² or normalized between different experiments by the largest lesion and expressed as % of the largest lesion, each boxplot represents at least 15 different leaves.

Next, we assessed whether the deletion of *BcPNL1* could affect pathogenicity on high and low N detached leaves. The *ΔBcpnl1* mutants exhibited a substantial reduction in pathogenicity on Columbia (Col-0) under both nitrogen nutrition conditions after 2 and 3 dpi compared to B05.10 infection (Fig. 3A, B). qPCR quantification of fungal DNA during the infection confirmed the reduced pathogenicity of these mutants (Supplementary Fig. 6). Furthermore, we observed an increased susceptibility of high N detache,d leaves infected either by B05.10 or *ΔBcpnl1* mutants (Fig. 3B) suggesting that BcPNL1 pectinolytic activity is not sufficient to explain the impact of host nitrogen nutrition on disease severity. Moreover, we observed a partial restoration of the pathogenicity with both complemented strains on Col-0 *A. thaliana* detached leaves (Fig. 3C) which is consistent with their *BcPNL1* expression level (Fig. 3C). To

characterize the origin of the pathogenicity defect, we analyzed the germination and penetration abilities of the *ΔBcpnl1* mutants on onion epidermis (Fig. 3D). Interestingly, the *ΔBcpnl1* mutants exhibit a delay in spore germination compared to the wild-type B05.10 strain, although penetration was still observed at the 24 hpi time point as for the wild-type.

We then confirmed that the role of BcPNL1 in pathogenicity is not restricted to Col-0 *A. thaliana* infection since detached leaves of tomato plants and Wassilewskija (Ws) *A. thaliana* plants infected with the *ΔBcpnl1* mutants also presented smaller lesions compared to B05.10 (Fig. 3E–G). Lastly, we compared the pathogenicity on Col-0 detached leaves of *ΔBcpnl1*, *ΔBcpg1* and *ΔBcpme1ΔBcpme2* and observed that the impact of *BcPNL1* deletion was more dramatic than that of *BcPG1* (Fig. 3H), which is consistent with the fact that the digestion of highly methyl-acetylesterified pectins by *BcPNL1* occurs earlier (i.e pre- and penetration stage) to BcPG1 activity, of which the corresponding gene reaches maximum expression upon lesion spreading[21]. In contrary to the increased pathogenicity observed previously with the *ΔBcpme1ΔBcpme2* mutant on Ws detached leaves[17], we found here a WT-like pathogenicity level on Col-0 as observed previously on grapevine and tomato leaves[40] indicating some host specificity in the role of *B. cinerea* pectin methylesterases.

### BcPNL1 is involved in the reduced cell death and oxidative burst responses during A. thaliana/ B. cinerea interaction

To characterize the plant response to the hypovirulent *ΔBcpnl1* mutants, we analyzed the oxidative burst by measuring $H_2O_2$ production using a 3,3′-diaminobenzidine (DAB) staining at an early time point after inoculation. At 1 dpi, high and low N Col-0 leaves infected with B05.10 strain exhibited a strong DAB staining at the inoculation site and its vicinity, while leaves infected with the *ΔBcpnl1* mutants showed only slight staining at the inoculation site (Fig. 4). We also assessed plant cell death upon infection using trypan blue (TB) staining. At the same time point (1 dpi), TB staining revealed fungal growth as well as small, darker spots indicating plant cell death, but this was observed only for the B05.10 strain at high and low N (Fig. 4B, C). These results indicate that at this early time point, the plant had strongly responded to the pathogenic B05.10 strain and lesser to the mutants.

### ΔBcpnl1 mutants produce less PNL and PG products at high N and low N

Next, we investigated the impact of *BcPNL1* deletion on cell wall degradation. The analysis of pectin degradation products in the presence of the *ΔBcpnl1* mutants at 15 hpi revealed on one hand a drastically reduced production of PNL-derived OGs compared to the WT strain, with both high N and low N detached leaves (Fig. 5A), confirming the importance of BcPNL1 on the total fungal PNL activity. On the second hand, PG products were also strongly reduced upon infection with the mutants, indicating that the PNL activity is essential upon infection to regulate PG and PME activity. Next, we performed an OG analysis on high N or low N detached leaves infected by *ΔBcpnl1.1c1* and the *ΔBcpnl1.1c2* strains at 18 hpi. We chose 18 hpi in order to assess in the meantime if any compensation mechanism would have been triggered in *ΔBcpnl1.1* mutant at a later time point. At this later time point than our previous analysis (Fig. 1A), high DP OGs couldn't be detected but we could still observe that the production of OGs was partially restored with the complemented strains compared to the *ΔBcpnl1.1* mutant (Fig. 5B). No clear conclusion could be formulated regarding PNL products for which only small DP were detected suggesting that, at this time point, most of the PNL products were already degraded. However, we observed with the *ΔBcpnl1.1c* strains the restoration of PG products accumulation for DP3 and DP4 in both N conditions and DP5 in low N that confirms that PNL activity is essential prior to PG and PME activity.

In order to assess if BcPNL1 activity could regulate the other pectinases at the transcriptional level or if the decrease of PG products only results from

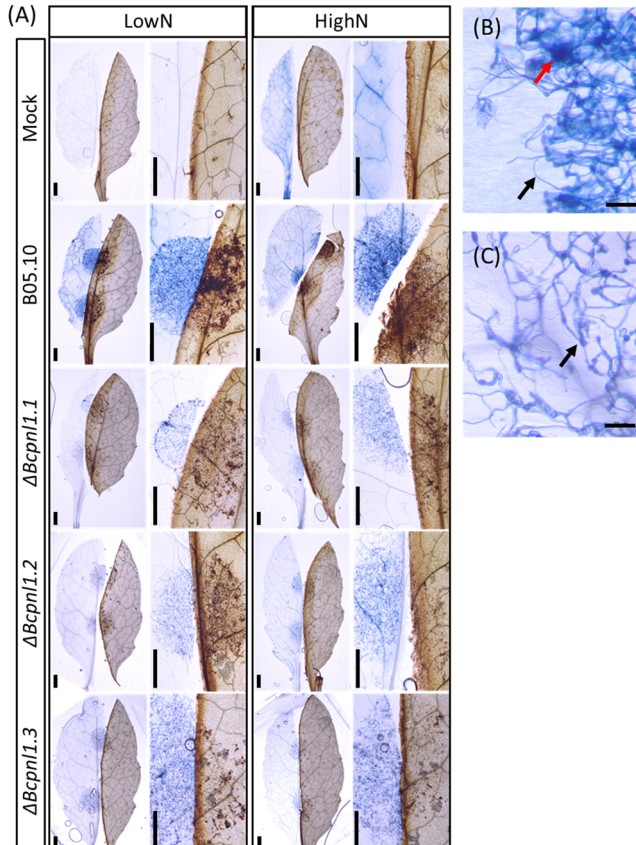

**Fig. 4 | Cell death and oxidative burst reaction during infection with WT and mutant *B. cinerea* strains.** Detached leaves of *A. thaliana* were infected with drops either only with ½ PDB (Mock) or drops containing spores of WT or *ΔBcpnl1* mutant strains of *B. cinerea*. **A** One day after inoculation, treated leaves were cut in half and each half was immediately colored with Trypan Bleu (TB) for cell death visualisation (on the left of each image) or with 3,3′-diaminobenzidine (DAB) for oxidative burst visualisation (on the right in each image). Scale bars = 500 μm. Zoom of a WT (**B**) or mutant (**C**) infected leave colored with TB. Black arrows show the fungus and the red arrow shows a spot of induced necrosis. Scale bar = 50 μm. Images were taken with a ZEISS Axio Zoom V16.

the inability of PMEs or PGs to have access to their substrate without prior PNL activity, we assessed the level of expression of pectinase genes in *ΔBcpnl1* mutants at 1 dpi (Fig. 5C). Interestingly, at 1dpi all the tested pectinase genes were downregulated in the *ΔBcpnl1* mutants compared to the WT strain in both N conditions. At 2 dpi, although less pronounced, we could still observe downregulation of the three *BcPNLs*, *BcPG4*, *BcPG5*, and *BcPG6* in the mutants whereas it was more variable for the *BcPMEs* and *BcPG1*. Interestingly, *BcPG2* and more clearly *BcPG3* were upregulated in the *ΔBcpnl1* mutants suggesting a delayed compensation mechanism to degrade pectin.

Taken together, these results suggest a lower aggressiveness of the *ΔBcpnl1* mutants compared to B05.10 at high and low N, consistent with the findings from the pathogenicity tests, the cell death and oxidative burst responses and the OGs analysis.

### The impact of host nitrogen nutrition on plant JA-related gene expression is partially altered during infection with the ΔBcpnl1 mutants

Next, we investigated the expression of plant defense-related genes during the infection (Fig. 6). At 1 dpi, the expression of plant defense-related genes, such as the JA marker gene *AtPDF1.2a*, the chitinase *AtPR4*, and *AtPAD3* coding for the final enzyme involved in camalexin biosynthesis[41], was

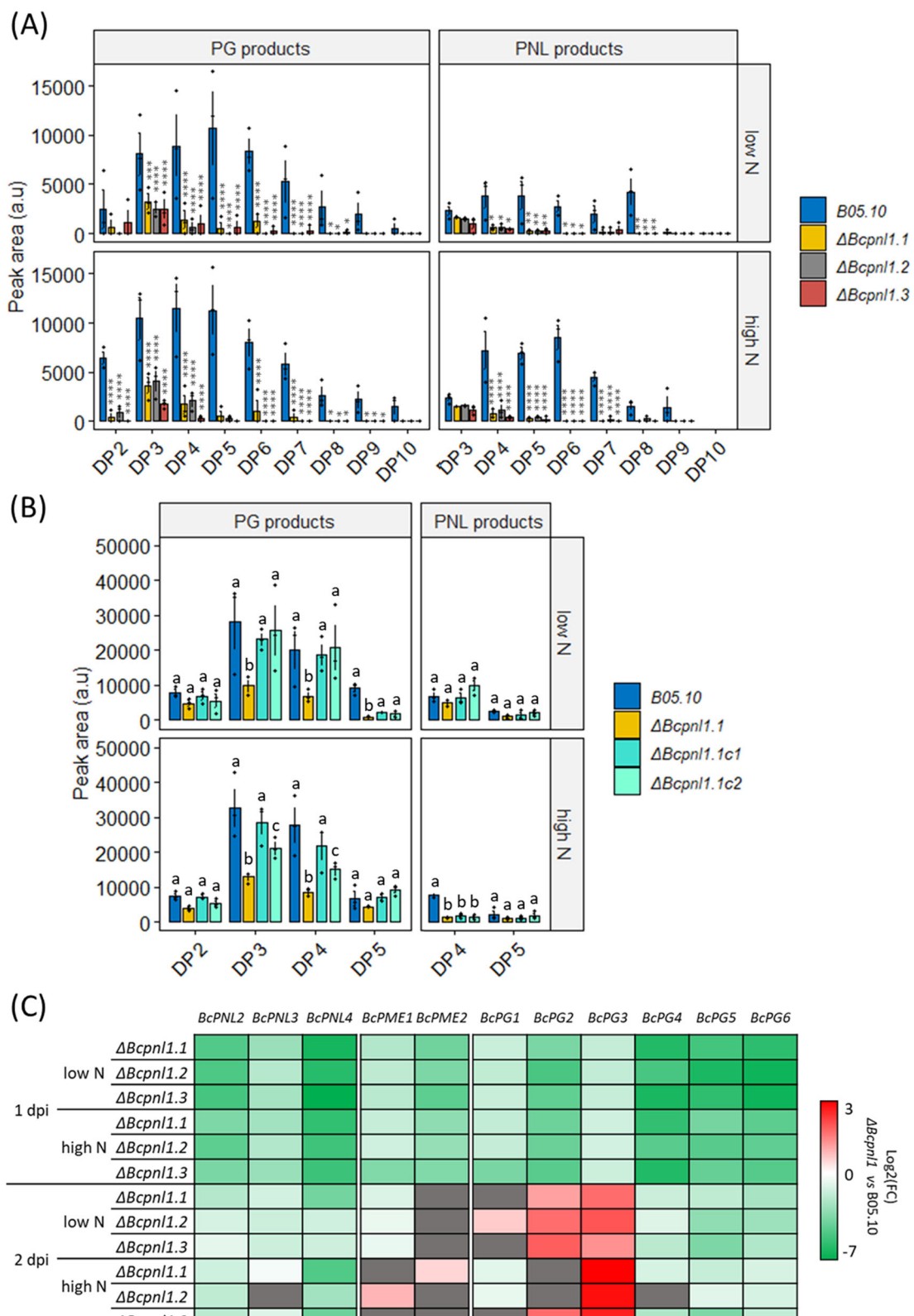

**Fig. 5 | Impact of *BcPNL1* mutation on *B. cinerea* pectinolytic capacities. A** LC-MS analysis of OG production from leaves of *A. thaliana* plants (cultivated at high or low N) incubated for 15 h in liquid medium supplemented with spores of the wild-type and mutant strains of *B. cinerea*. A two-way ANOVA test was performed followed by a Fisher's LSD test between wild type and mutant strains: *$p < 0.05$, **$p < 0.01$, ***$p < 0.001$. **B** LC-MS analysis of OG production from leaves of *A. thaliana* plants (cultivated at high or low N) incubated for 18 h in liquid medium supplemented with spores of the WT, mutant and complemented strains of *B. cinerea*. Letters represent the results of a two-way ANOVA test followed by a Fisher's LSD test between strain for each DP group ($p > 0.05$). DP: degree of polymerisation. **C** Heatmap representing the log2 fold change of fungal pectinase transcript accumulation between the B05.10 reference strain and the *ΔBcpnl1* mutants during infection in high and low N conditions at 1 dpi and 2 dpi on detached leaves. Fungal gene targets were normalized with *BcActA* and similar results were obtained using *BcUBI* for normalization. Three independent experiments were conducted with similar results. Data are means of triplicates (±SE). Differences that were not significant according to an unpaired t-test are represented in gray: $p < 0.05$.

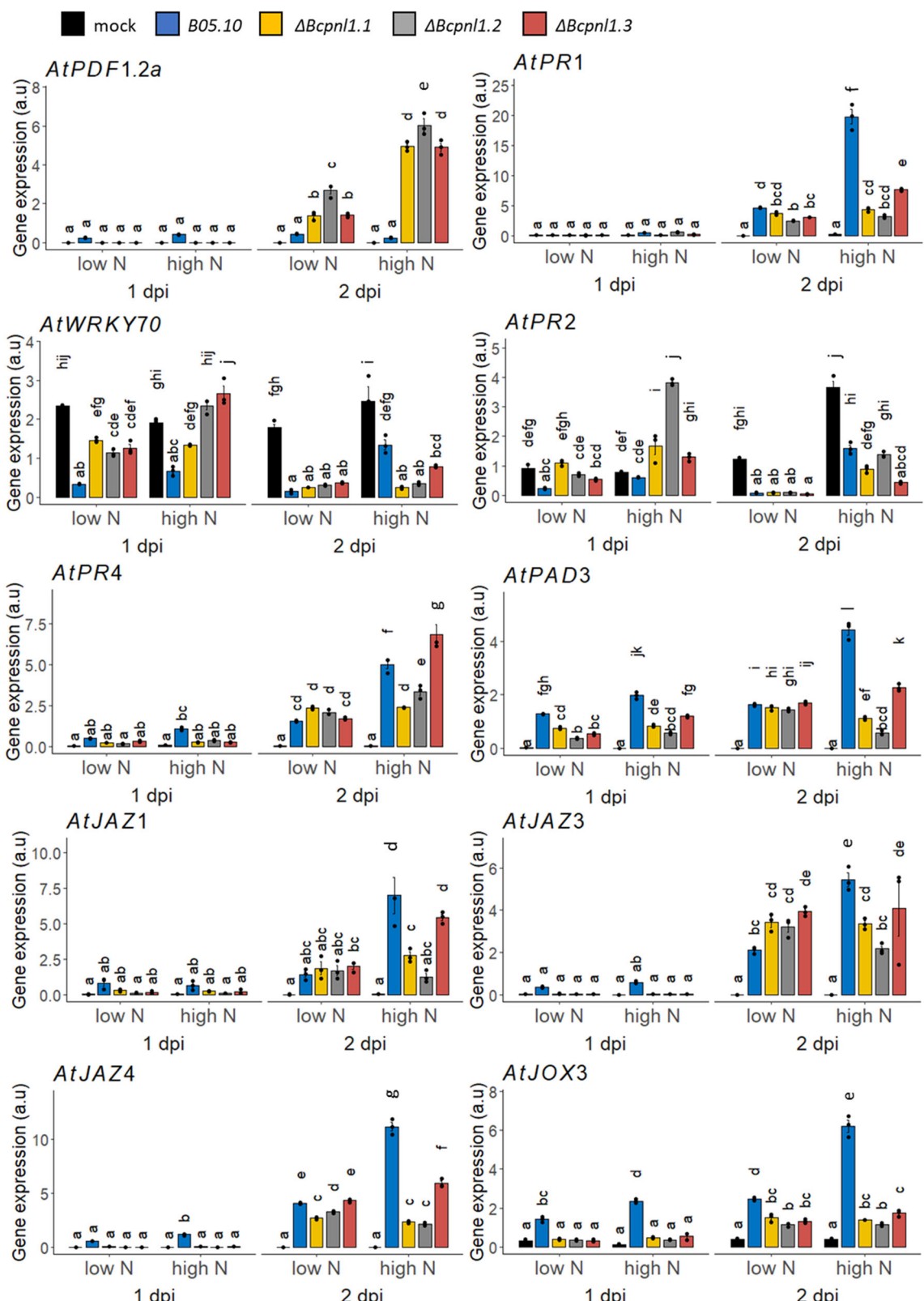

**Fig. 6 | Differences in plant defense gene expression during infection of detached leaves from high and low N *A. thaliana* plants with the B05.10 reference strain and the *ΔBcpnl1* mutant strains.** Transcript accumulation of plant defense related genes obtained by RT-qPCR from detached leaves after 1 and 2 dpi with the WT and *ΔBcpnl1* mutant strains in high N and low N conditions. Letters are the result of an ANOVA followed by a fisher LSD test ($p < 0.05$). Plant gene targets were normalized with *AtUBI4*. Data are expressed as mean normalized expression in arbitrary units (a.u.) and are the means of triplicates (±SE). Similar results were obtained using *AtAPT1* for normalization. Three independent experiments were conducted with similar results.

decreased in detached leaves infected with the *ΔBcpnl1* mutants compared to the WT strain. However, at 2 dpi, transcript levels in detached leaves infected with the *ΔBcpnl1* mutants increased in all cases compared to 1 dpi, particularly under high N conditions. In this scenario, *AtPDF1.2a* expression was notably more elevated upon infection with the *ΔBcpnl1* mutants than with the WT. These initial data suggest that the activation of defense mechanisms, particularly through JA production, is delayed in presence of *ΔBcpnl1* mutants and modulated by nitrate content.

We also observed that the expression of *AtJOX3* and the transcription factors *AtJAZ1, 3*, and *4* was down-regulated in detached leaves infected with the mutants compared to the WT at 1 dpi under all conditions (Fig. 6). Interestingly, this down-regulation persisted at 2 dpi, but only under high N conditions. On the other hand, the expression of the *AtPR1* gene was very low at 1 dpi across all conditions, but an increase was observed particularly with the WT strain and under high N concentration. With the mutants, however, this response was diminished under high N, suggesting a decrease in antimicrobial activity in these conditions[42].

The expression profile of the *AtPR2* gene, coding for a β-1,3-glucanase with antifungal properties also involved in callose deposition and SA-dependent defense responses[43,44], indicates that the presence of *ΔBcpnl1* mutants on detached leaves triggered a plant response linked to SA at 1 dpi in high N which is absent in mock or upon inoculation with the WT. At 2 dpi, *AtPR2* expression was induced in the high N mock compared to 1 dpi, which may be an effect of using detached leaves. By contrast, *AtPR2* expression was downregulated upon infection compared to the mock in a similar way with the WT and the mutants. (Fig. 6). Finally, the expression of the *AtWRY70* gene, which is involved in SA and JA/ethylene signaling pathways[36,37], was repressed upon infection compared to the mock at 1 dpi suggesting the presence of JA. This gene repression was notably higher with the WT compared to the *ΔBcpnl1* mutants in both N regimes. At 2 dpi, *AtWRY70* expression was still repressed by the infection but interestingly the repression was higher with *ΔBcpnl1* mutants compared to the WT under high N conditions which might reflect a delay in *AtWRKY70* repression with *ΔBcpnl1* mutants. This indicates that SA and JA/ET signaling is modified during the kinetics of *B. cinerea* infection to tend to produce JA and this process is influenced by N levels and the absence of *BcPNL1*.

All these differences in gene expression profiles in presence of the WT strain or the mutants suggest a delay in the establishment of the plant's defense mechanisms in the presence of *ΔBcpnl1* mutants, which is more pronounced under high N conditions. This weak repression of the JA biosynthesis pathway is correlated to the absence of high DP unsaturated OGs in *ΔBcpnl1* mutants under high N conditions as proposed in the model presented in Fig. 7.

## Discussion

*Botrytis cinerea*, like all pectin-degrading fungal pathogens, relies on its pectinases to break the cell wall barrier and release nutrients that support fungal growth and development. However, the impact of abiotic factors such as nitrogen nutrition on this cell wall degradation process, as well as the role of PNLs as pathogenicity factors in *B. cinerea* pathogenesis, had never been reported. In this study, we were able to point towards the involvement of BcPNL1, encoded by the most expressed *PNL* genes in the early phase of infection (Fig. 1 and Supplementary Fig. 1) and from which the *in* vitro PNL activity was previously confirmed[17] in orchestrating part of the interaction between host nitrate nutrition and disease severity.

Thanks to the recently described CRISPR-Cas9-mediated gene replacement in *B. cinerea*[38], we have obtained and then characterized *ΔBcpnl1* mutants (Fig. 2). *In* vitro, the mutants presented normal growth with glucose or highly methylesterified pectin as the sole carbon source (Supplementary Fig. 5). Similarly, *Δpnl1* mutants of *Penicillium digitatum* displayed no growth defects *in* vitro[45]. However, it would be interesting to test the growth of the mutants with more or less acetylated pectins given the preference of BcPNL1 for methyl-acetylated pectins (Supplementary Fig. 3). Interestingly, on onion epidermis we observed a delayed spore germination

of *ΔBcpnl1* mutants (Figs. 3D, 7A). In the transcriptome analysis of germinating spores on a plant mimicking surface, Leroch et al. (2013)[15] showed that *BcPNL1* expression is strongly induced after only one hour of incubation in agreement with our finding that BcPNL1 is the most expressed PNL at 1 dpi (Supplementary Figs. 1 and 2). These observations suggest that *BcPNL1* might be implicated in early events promoting germination upon perception of the plant surface.

In other phytopathogens such as *Pectobacterium carotovorum*, *P. digitatum* and *Verticillium dahlia*, PNL activities represent important determinants of pathogenicity[45–47]. Concerning *B. cinerea*, to our knowledge no study had clearly investigated the role of *BcPNL* genes. On detached leaves, we observed a reduced pathogenicity of *ΔBcpnl1* mutants restored in the complemented strains (Fig. 3) demonstrating that PNL activity is an important determinant of *B. cinerea* pathogenicity too. Furthermore, the reduced pathogenicity of the *ΔBcpnl1* mutants on different host plants also suggests that this pathogenicity factor is not linked to host specificity. Supporting this hypothesis, Cotoras & Silva (2005) observed in WT *B. cinerea* strains isolated from tomato and grape a similar level of secretion of PNLs. In other fungal pathogens such as *V. dahlia* and *P. digitatum*, mutants of PNLs were also hypovirulent on their respective hosts[45,46,48,49]. Authors not only mention their role in promoting penetration through cell wall degradation but also their necrotizing activity. In our study, according to trypan blue colorations, initial necrosis of the plant cells seemed to be reduced in presence of the *ΔBcpnl1* mutants (Fig. 4). Movahedi & Heale (1990) showed that a PNL purified from *B. cinerea* germinating spores had a cell death activity on carrot cell culture and was able to produce elicitors from cell wall components. Since *BcPNL1* was the only *PNL* gene that was strongly expressed in the early interaction, it might be the same enzyme that the authors purified and characterized as a necrosis-inducing protein. Finally, we observed that the increased susceptibility of high N plants to *B. cinerea* was still observed with *ΔBcpnl1* mutants suggesting that BcPNL1 pectinolytic activity alone is not sufficient to explain the impact of host nitrogen nutrition on disease severity.

We also showed that more PNL products were produced in high N plants infected with the B05.10 wild-type strain of *B. cinerea*, and this higher production correlates with a higher up-regulation of *AtJOX3* and *AtJAZs* (Fig. 1). These genes are involved in repression of the JA pathway which is strongly implicated in efficient defense strategy against *B. cinerea*[50,51] and are induced by long methyl esterified PNL products, accumulated upon plant infection with the *ΔBcpme1ΔBcpme2* mutant, hypervirulent on *A. thaliana*. With the *ΔBcpme1ΔBcpme2* mutant, these high DP (>4) PNL products, which cannot be further degraded by PGs induce an up-regulation of genes involved in JA pathway repression such as *AtJOX3* and *AtJAZs*[17]. Therefore, to explain why high DP PNL products were accumulated in high N infected plants, we assessed if the host nitrogen nutrition could impact the transcriptional regulation of *BcPNLs* and/or *BcPMEs*. However, we did not observe any significant up-regulation of these genes in high N condition that would have explained the accumulation of high DP PNLs OGs (Supplementary Fig. 2). Interestingly we found that pectin acetylesterification is required for PNL activity and that more methyl-acetylesterified OGs were produced upon inoculation of high N detached leaves in liquid compared to low N (Supplementary Fig. 3). Consequently, although we cannot rule out the possibility of post-transcriptional or post-translational regulation of PNLs, our results suggest that the increased release of high DP PNLs OGs in the high N condition is due to a higher level of methyl-acetylesterified regions allowing PNL activity as proposed in the model (Fig. 7). It was shown in *A. thaliana* that an acetyl-transferase mutant and a transgenic line producing fungal PAEs with reduced pectin acetylation level are more resistant to *B. cinerea* infection[20,52]. Thereby, a control of PNL activity by the pectin acetylation level could contribute to its impact on disease severity.

Consistently, in the *ΔBcpnl1* mutants, pectin degradation capacities were strongly affected both at high and low N. Far less PNL-derived cell wall

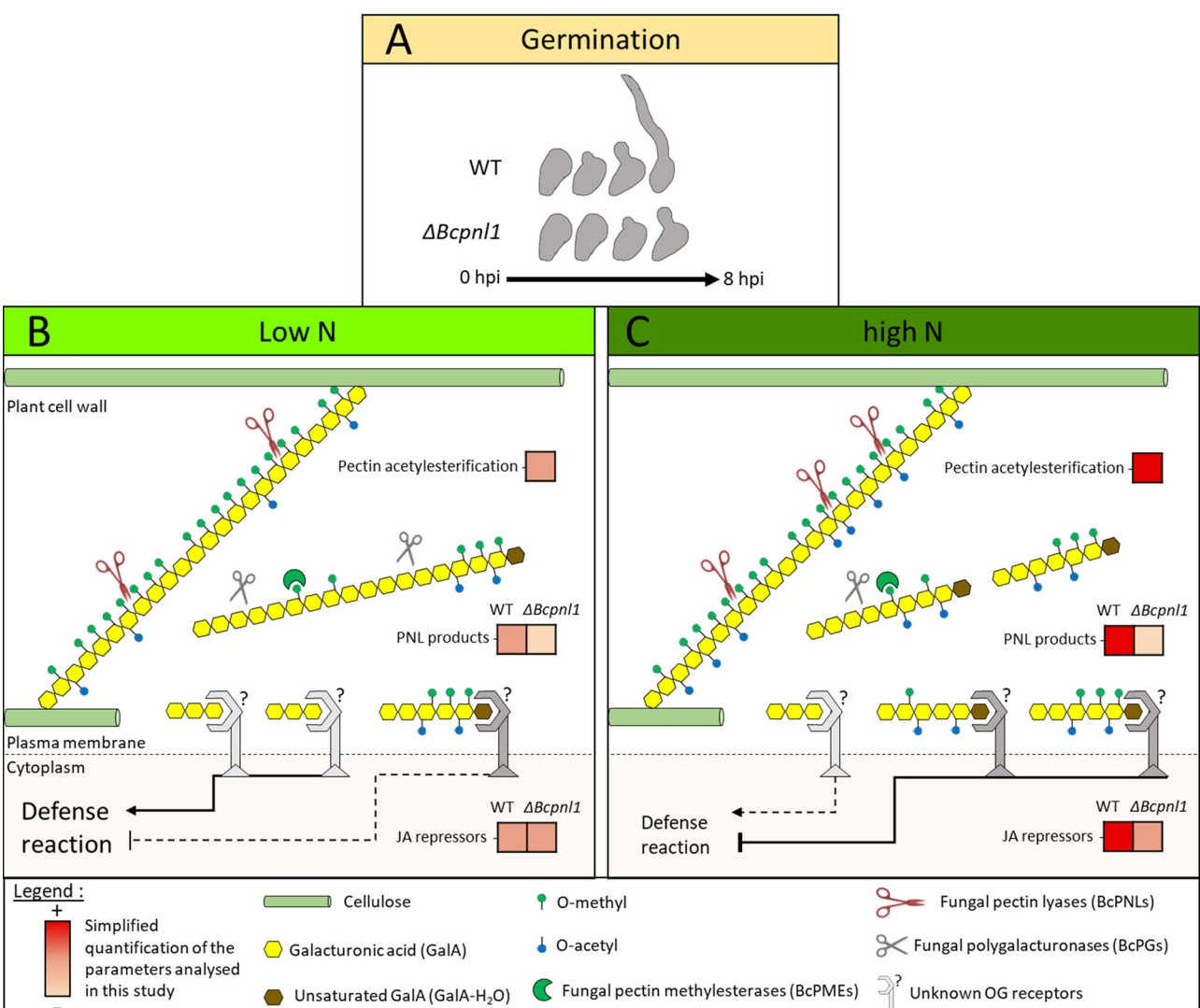

**Fig. 7 | Proposed model placing BcPNL1 in the interplay between plant nitrogen nutrition and disease susceptibility. A** The pectin lyase BcPNL1 is involved in early germination events on the plant surface. **B**, **C** Plant nitrogen nutrition affects early degradation of the plant cell wall by BcPNL1 probably because of differences in pectin acetylation levels which is a key factor for BcPNL1 activity. In this model, the stronger acetylation level of pectins in high N leads to an increased BcPNL1 activity and the accumulation of high DP methyl-acetylated and unsaturated OGs. This correlates on the plant side with earlier and stronger activation of JA repressor genes in high N, absent upon infection with the ΔBcpnl1 mutants confirming the role of BcPNL1-derived OGs in inhibiting plant defenses through unknown receptors.

products were detected upon infection with the ΔBcpnl1 mutants (Fig. 5) which corroborates the absence of functional compensation observed with the other pectin lyases at the transcriptional level (Fig. 5C). Only BcPG3 was strongly upregulated in the mutants at 2 dpi and BcPG2 to a lesser extent although they were downregulated at 1 dpi as all the tested pectinase genes and probably because of a delayed development of the fungus. Interestingly, BcPG3 was shown to have a better activity on low methylesterified pectin (DM 7%) compared to polygalacturonic acid[22] which can partly explain why it is compensating the BcPNL1 deletion at the transcriptional level and not the others. Indeed, it is known that pectinases are regulated at the transcriptional level by pectin and pectic components[18], thus, BcPNL1 products released upon degradation might induce genes coding for enzymes able to achieve the next degradation step. Overall, deletion of BcPNL1 seemed to modify deeply the kinetics of pectin degradation and OG turnover during infection which can explain the delayed pathogenicity of the mutants at high and low N.

During host-pathogen interactions, OGs produced by pectinases are important DAMPs which modulate immunity in plants[17,24,25]. In this study, the pectin lyase mutant ΔBcpnl1 impaired in pectin degradation presented a germination delay associated with a delay in plant defense induction

(Figs. 3D, 6, 7). Interestingly, at a later stage (2 dpi) the plant defense response, particularly the JA defense pathway, was stronger with the mutant pointing out its incapacity to repress early JA induction (Figs. 6, 7). This delayed hyper-induction of plant defenses caused by BcPNL1 deletion was exacerbated in high N conditions where the JA pathway repression is stronger in the WT and the BcPNL1 derived OGs more accumulated. These data underline on one hand, that high DP unsaturated OG produced by BcPNL1 are capable of suppressing plant JA-mediated defense activation and on the other hand that these OGs have a role in the nitrogen-dependent down-regulation of plant defense. In addition, using a large collection of fungal isolates, it was found that the level of BcPNL1 expression is positively correlated to the disease symptoms on A. thaliana Col-0 but not with the npr1 and coi-1 mutant lines affected in the SA and JA pathway respectively[53] reinforcing our finding that BcPNL1 is involved in plant defense modulation. It would be of interest to investigate the exact mechanism by which those specific BcPNL1-derived OGs are perceived and trigger the downstream plant defense repression. A recent study showed that the previously designated OG receptors that belong to the wall-associated kinase family (WAK, 5 members) are in fact dispensable for the OG-mediated immune response[54] indicating that there is still much to discover on OG signalling.

To conclude, we were able in this work to describe the pectin lyase BcPNL1 which, to the best of our knowledge, is an unreported pathogenicity factor of *B. cinerea* crucial for pectin degradation. Its activity appears to be affected by pectin acetylation changes induced by host nitrogen nutrition resulting in a greater production of high DP PNL-derived OGs at high N which partially hijack the DAMP-mediated immunity.

## Methods

### Fungal and plant material

*B. cinerea* strain B05.10 collected from *Vitis* in Germany[55] was used throughout this work as the WT reference strain. Strains of *B. cinerea* were routinely cultured on Potato dextrose agar (PDA, Difco) or Malt Agar (1% malt extract, 1.2% agar) at 23 °C under constant light. For fungal growth rate measurements, we used Czapeck minimal media containing 2.55 g.l$^{-1}$ NaNO$_3$, 0.5 g.l$^{-1}$ KCl, 0.5 g.l$^{-1}$ MgSO$_4$.7H2O, 10 mg.l$^{-1}$ FeSO$_4$.7H$_2$O, 1 g.l$^{-1}$ K$_2$HPO$_4$, 10 mg.l$^{-1}$ Na$_2$MoO$_4$.2H$_2$O and either 30 g.l$^{-1}$ glucose or 10 g.l$^{-1}$ of highly methylesterified pectin from citrus peel (DM > 70%, P9561, Sigma-Aldrich) as the sole carbon source. *A. thaliana* ecotypes Col-0 and Ws-0 were obtained from the INRAE-Versailles collection. For all experiments, plants were cultured in growth chambers (65% relative humidity, 8 h photoperiod, 21 °C) for 6 weeks on nonsterile sand either supplied with a nutrient solution containing 0.5 mM NO$_3^-$ (Low N) or 10 mM (High N). *Solanum lycopersicum* plants (Monalbo) were cultured in the same conditions for 6 weeks but were only supplied with 10 mM NO$_3^-$ nutritive solution.

### Construction of the ΔBcpnl1 mutants and complemented strains

*ΔBcpnl1* mutants and complemented strains were obtained by using CRISPR-Cas9 mediated homologous recombination (HR) (Fig. 2)[38]. Briefly, fresh spores of the B05.10 strain (1.10$^8$ conidia) were cultured overnight in 100 ml of HA liquid medium (1% malt extract, 0.4% glucose, 0.4% yeast extract, pH 5.5). The obtained fungal biomass was washed twice in KCl/NaPi buffer (0.6 M KCl, 100 mM sodium phosphate pH 5.8) and then incubated in KCl/NaPi buffer with 1.35% (w/v) protoplasting enzymes (Glucanex) 1 h at 28°C on a rotary shaker at 60 rpm. Protoplast were then washed twice with TMS buffer (1 M sorbitol, 10 mM MOPS, pH 6.3) and concentrated in TMSC buffer (TMS + 50 mM CaCl$_2$) at the desired concentration (2.10$^7$ conidia.100 µl$^{-1}$). PEG-mediated transformation was achieved by addition of PEG solution (0.6 g ml$^{-1}$ PEG 3350, 1 M sorbitol, 10 mM MOPS, pH 6.3) to the protoplasts previously incubated for 10 min on ice with the ribonucloprotein (RNP) complex and the DNA repair template (10 µg). The RNP complex was obtained by incubating for 30 min at 37 °C in 15 µl of EngesCas9 specific buffer (NEB), 6 µg of the commercial EngenCas9 (NEB) with 2 µg of the freshly synthetized SgRNA1 obtained with the HiScribe™ T7 High Yield RNA Synthesis Kit (NEB). DNA repair templates containing the fenhexamid resistance cassette surrounded by 60 bp of the target gene for HR were obtained by PCR. For the *ΔBcpnl1* mutants, primer X1 and X2 were used on the pTEL-fenh plasmid kindly provided by Pr. Matthias Hahn (Kaiserslautern, Germany). For the complemented strain c1, primers X3 and X4 were used to amplify the coding sequence of *PNL1* together with the native promoter (1.5 kb) and the terminator (0.5 kb*)*. For the complemented strains c2 the oliC promoter and the gluc terminator were amplified from the pndh-ogg plasmid[56], kindly provided by Muriel Viaud (INRAE, Saclay), using the primer pairs X5-X6 and X7-X8 respectively whereas the *PNL1* coding sequence was amplify with X9 and X10. To amplify the hygr cassette, the X11 and X12 primers were used on the pndh-ogg. Finally, the two or four fragments respectively were then assembled by overlapping tail PCR using X13 and X14 to form the final fragment carrying the 60 bp for HR in an intergenic region of chromosome 3[39]. The protoplasts were then plated on SH agar medium (0.6 M sucrose, 5 mM Tris-HCl pH 6.5, 1 mM (NH$_4$) H$_2$PO$_4$, 9 g.l$^{-1}$ bacto agar) supplemented with 1 µg.ml$^{-1}$ fenhexamid or 40 µg.ml$^{-1}$ hygromycin B for regeneration. Resistant transformants were then subcultured with the selection pressure on HA agar (HA, 1.5% agar) and once again on PDA and MA for stock solution of spores and DNA extraction respectively. Finally,

correct homokaryotic integration was verified by PCR (Supplementary Fig. 4A and C) on genomic DNA extracted[57]. Unique integration of the cassette in the mutants was then controlled by Southern blot (Supplementary Fig. 4B) using DIG-High Prime DNA Labeling and Detection Starter Kit I (Sigma). Primers used for these experiments are listed in Supplementary Table 1. Selection of the SgRNA protospacer sequences was done with the online CRISPOR tool[58].

### Detached leaves infection experiments

For all infection experiments, spores of each strain were harvested from 2 weeks-old PDA cultures. For pathogenicity assays, RT-qPCR experiments and colorations, detached leaves were inoculated on their upper surface with 20 µl drops containing 5.10$^5$ conidia. ml$^{-1}$ in ½ Potato dextrose broth (PDB, Difco). Leaves were then kept in a Petri dish under high humidity conditions and daylight at 23°C. Fungal DNA quantification in infected leaves was performed according to[59].

### DAB and BT coloration experiments

For coloration assays, leaves were infected as described above and cut in half 24 h after inoculation prior to coloration. Detection of H$_2$O$_2$ production was achieved by 3,3'-diaminobenzidine (DAB) staining. Briefly, samples were immersed in an aqueous solution containing 1 mg.ml$^{-1}$ DAB (pH 3.6, Sigma) and incubated overnight at RT on a rotary shaker. To discolor the leaves, chlorophyll was extracted by incubation in 75% ethanol. To detect cell death, Lactophenol Trypan Blue (TB, Sigma) was performed[60]. Pictures were taken on a ZEISS Axio Zoom V16.

### Gene expression analysis

For RT-qPCR, in each condition total RNA was extracted from a pool of 8 leaves after 1 and 2 dpi and extracted using TRIzol reagent (Invitrogen). Three independent biological replicates were analyzed. Reverse transcription was performed using an oligo-dT20 for a primer and Superscript II RNaseH-reverse transcriptase (Invitrogen). PCR was performed on a Applied Biosystems™ QuantStudio™ 5 Real-Time PCR System with SYBR Green PCR MasterMix (Eurogentec). Each reaction was performed on a 1:20 dilution of the cDNA, synthesized as described above, in a total reaction of 5 µl. Gene expression values were normalized to expression of the *A. thaliana AtAPT1* and *AtUBI4* genes or *B. cinerea BcACTA* and *BcUBI*[61]. We obtained similar results with both genes, therefore results with only one reference gene are shown. Specific primer sets are given in Supplementary Table 2.

### Analysis of OG production

For OG production experiments, triplicates of 10 detached leaves were directly immersed in 35 mL of a *B. cinerea* spores suspension containing 3.10$^5$ conidia.ml$^{-1}$ in Gamborg's B5 basal medium, 2% fructose and 10 mM phosphate buffer pH 6.4[17]. Samples were incubated at 23 °C on a rotary shaker at 80 rpm. After 12, 15, or 18 h of incubation of the leaves in the suspension of spores an equal volume of ethanol 96% was added in order to precipitate the largest molecules. For assays performed on pectins, GAT103 and GAT300 were used as methyl- and methyl/acetylesterified pectins, respectively, https://www.elicityl-oligotech.com/ and incubated overnight in Gamborg medium containing *B. cinerea* spores at 3.10$^5$ spores.ml$^{-1}$. For the analysis of OG production, samples were diluted to 1 mg.ml$^{-1}$ in 50 mM ammonium formate containing 0.1% formic acid. Chromatographic separation utilized the ACQUITY UPLC Protein BEH SEC Column (125 Å, 1.7 µm, 4.6 mm × 300 mm; Waters Corporation, Milford, MA, USA) with elution in 50 mM ammonium formate and 0.1% formic acid at a flow rate of 400 l.min$^{-1}$ and a column oven temperature of 40°C and was made using an HPLC system (UltiMate 3000 RS HPLC system, Thermo Scientific, Waltham, MA, USA). The system was coupled to an Impact II Ultra-High Resolution Qq-Time-Of-Flight (UHR-QqTOF) spectrometer (Bruker Daltonics, Bremen, Germany) equipped with an electrospray ionisation (ESI) source in

negative mode with the end plate offset set voltage to 500 V, capillary voltage to 4000 V, nebulizer to 40 psi, dry gas to 8 l/min and dry temperature of 180 °C. The Compass 1.8 software (Bruker Daltonics) was used to acquire the data. Peak annotation was performed based on accurate mass annotation, isotopic pattern, and MS/MS analysis[17].

## Statistics and Reproducibility

All statistical analyses were done using the RStudio software (2024.09.1, Build 394). All bar graphs represent the mean value and the standard error as errors bars, points represent individual sample values. In the box plots median and interquartile range are indicated, with median shown as black line inside the boxes and interquartile range shown as the length of the boxes. For oligoprofiling data, an ANOVA followed by a Fisher's LSD post hoc test were conducted for multiple comparisons. For data presented with the aim of comparing low N and high N conditions or WT and mutants, a two-sample unpaired T-test was conducted, if both comparisons were done at the same time a two-way ANOVA (or a three-way ANOVA if the time factor was added) followed by a Fisher's LSD post hoc test was used. Differences were considered significant when the p value was lower than 0.05. All gene expression analyses were done with three technical replicates and one of the three independent experiment showing similar data was presented. For oligoprofiling data, three or four technical replicates and one of at least three independent experiment showing similar data was presented. For pathogenicity tests, at least 15 technical replicates each representing one inoculated detached leaf were done for each individual experiment repeated at least three times.

## Reporting summary

Further information on research design is available in the Nature Portfolio Reporting Summary linked to this article.

## Data availability

All data supporting the findings of this study are provided within the paper and its Supplementary Information. Source data for the charts and graphs presented in the main figures are available as Supplementary Data. The raw LC–MS datasets can be accessed at https://doi.org/10.6084/m9.figshare.28290053.v1. Materials and all other data are available from the corresponding authors upon reasonable request.

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

## Acknowledgements

This work has benefited from the support of IJPB's Plant Observatory technological platforms and from the following funding French National Research Agency ANR-22-CE43-0013-WALLDERIVE (A.V., S.V., and L.L.). M.F., A.D., and M.C.S. were awarded the Eqolysin Grant (Award Number: BAP2022_35). A.D. was awarded a grant from the doctoral school SEVE of U. P. Saclay. IJPB received support from Saclay Plant Sciences-SPS (ANR-17-EUR-0007).

## Author contributions

Conceptualization: A.D., A.V., S.V., M.C.S., and M.F. Methodology: A.D., A.V., M.C.S., M.F., L.L., and S.J. Investigation: A.D. and A.V. Visualization: A.D. and A.V. Supervision: M.F., M.C.S., and A.V. Writing—original draft: A.D. and A.V. Writing—review & editing: A.D., A.V., S.V., M.C.S., and M.F.

## Competing interests

The authors declare no competing interests.
