## [Transparent Peer Review file · Communications Biology]

Unravelling the Interplay of Nitrogen Nutrition and the Botrytis cinerea pectin lyase BcPNL1 in Modulating Arabidopsis thaliana Susceptibility

Corresponding Author: Dr Aline Voxeur

Version 0:

Reviewer comments:

Reviewer #1

(Remarks to the Author)

General remarks and main concerns:

In this manuscript, Davière et al. present the impact of the Botrytis cinerea pectin lyase BcPNL1 in modulating Arabidopsis thaliana susceptibility under 2 nitrogen nutrition conditions. The manuscript is clear and the presented results are relevant. But my main concern is about the notion of response to infection which is not always observed at the same time after infection, it raises questions. Ideally it would be interesting to observe the dynamics of the infection with time-course analyses. But otherwise, the authors should at least justify why some analyses are done 1 day after infection, others 2 days after and also 15 or 18 hours after infection for LC-MS analyses. Another important point for me is that the study of the response is done on detached and incubated leaves. How can this biological model be consistent with the response of the leaf attached to the plant, particularly when looking at hormonal signalling?

Minor concerns

L 91 "GalA4MeAc-H₂O" is an abbreviation, please give the full name

L 124 "Since SA and JA defence signalling pathways are mutually antagonistic, ..." : add a reference

L 178 Could you evaluate the number of spores to compare the quality of the mutant versus the WT?

L 204 Fig 3C What is the BcCUTA gene?

L 224 and Fig 5AC. The authors claim that "pectin degradation products in the presence of the Δ BcPnl1 mutants ... revealed a drastically reduced production of PNL derived OGs compared to the WT strain" but the data presented are expressed in peak area. It is best to use an absolute value when comparing a quantity, as each PG and NLP product standard may have different profiles.

Fig5B: At the top, what do the numbers mean?

Reviewer #2

(Remarks to the Author)

In this manuscript, the involvement of the pathogen pectin lyase (BcPNL1) in the nitrogen-induced susceptibility of the interaction between Arabidopsis and Botrytis cinerea, previously reported by the authors, is analyzed using a pathogen mutant (Δ BcPnl1). The authors also analyzed changes in host pectin metabolism during B. cinerea infection and suggest that PNL-derived oligogalacturonides, whose accumulation increases under high nitrogen conditions, reduce host resistance levels via induction of expression of repressors of jasmonic acid signaling. They then propose that this may be involved in nitrogen-induced susceptibility. The experimental results are clear and suggestive to a certain degree. However, there are some points that must be carefully considered in order to support the conclusions proposed in this paper.

Major points

1. The main data in this paper are considered to be the results of the analysis using Δ BcPnl1.

Since $\Delta Bcpl1$ reduces virulence, there is no doubt that BcPNL1 has an important role as a pathogenesis factor in *B. cinerea*. However, as shown in Figure S2, due to the slow germination of spores in $\Delta Bcpl1$, the subsequent infection process is delayed compared to WT. It is highly possible that this timing discrepancy will also affect subsequent host responses. Unfortunately, each of the experiments shown was performed at a single time point and does not account for the staggered timing of infection. To solve this problem, experiments using $\Delta Bcpl1$ should show detailed changes in the infection process over time, and then changes in gene expression should be shown in time course.

2. The authors concluded that high nitrogen affected the host pectin structure, since the accumulation of PNL products increased even though the expression of PNL in *B. cinerea* was not increased under high nitrogen conditions (lines 111-116, 329-332 lines). I agree with this. However, it is not clear exactly how the pectin structure of the host was altered. Additional experiments should be conducted or the experimental data and other findings should explain what the possibilities are.

3. The authors state in lines 184-187 and 364-366 that BcPNL1 alone cannot explain nitrogen-induced susceptibility. The authors claim that BcPNL1 is involved in nitrogen-induced sensitivity because JAZ gene expression is not significantly different between high and low nitrogen conditions in the mutant in Figure 7B, but the data are not shown in detail enough to make an objective determination. The involvement of BcPNL1 in nitrogen-induced sensitivity may be negligible, if any. If so, the title and abstract may be misleading to the reader.

Minor points.

1. Why was the analysis performed at 18hpi in Figure 5C?

2. Line 239-241. I don't know why the low PG product of $\Delta Bcpl1.1c2$ confirms that PNL is essential prior to PG activity.

Reviewer #3

(Remarks to the Author)

Few studies in plant pathology have considered all components of the disease triangle, i.e. pathogen virulence, host response and environmental effects. However, abiotic stresses certainly have a major influence on biotic stresses, which is still largely unknown. By investigating the effect of plant nitrogen fertilisation on disease development by the *Botrytis* fungus, this study shows that the degradation of plant pectin by the fungus is regulated as a function of plant nitrogen nutrition. As well as identifying a new virulence factor in this fungus, this study has the wider merit of drawing our attention to the need to consider nitrogen fertilisation and, more generally, the impact of the environment on the virulence of plant pathogenic fungi. Because of its innovative aspect, I am rather in favour of the publication of this article, but a revision seems to me to be necessary.

Abstract :

The abstract needs to be rewritten and would benefit from being more factual.

There is a lack of precision in "revealed a noteworthy impact of nitrogen availability ... on PNL1". Is this a positive or negative effect? I wouldn't mention the mutant in the same sentence as it confuses the information. The nature of the effect is then explained, but this introduces redundancy that is not expected in a summary. I don't think we can say that the enzymatic activity of PNL1 is affected. This implies that the effect is direct, but there is no specific PNL1 enzymatic assay to verify this. This can be observed indirectly by deleting the gene that encodes this activity and measuring OG degradation products in the mutant. These degradation products may originate from PNL1 as well as from other PNLs and there may be epistatic relationships between the genes in this family.

Introduction:

I did not find the review by Cheung et al. 2020 in the list of references. Please check again all references.

It says that *Botrytis* has a high degree of functional redundancy and compensation phenomena, but these characteristics do not seem to be specific to this fungus and do not apply to all genes. I would rather explain that among the virulence factors, some are encoded by multi-gene families such as CAZY, which are subject to functional redundancy and compensation phenomena that make them difficult to study in fungi.

I wonder if there isn't a misunderstanding about the role of chitin synthase activities such as *chs3a*. It is said that they activate plant defence, but according to the title of the article cited, it would be the absence of *chs3a* in a deletion mutant that would induce the response.

The paragraph between lines 52-70 explains the enzymatic arsenal of the fungus dedicated to the degradation of plant pectin. It's complex and I think a scheme as a supplementary figure would be helpful, especially to understand the differences between PNL and PL activities.

It is not specified how many genes are listed for each type of pectinolytic activity in *Botrytis*.

The paragraph between lines 71 and 91 explains what is known about the plant's perception of OGs and its immune response. Could we have some clarification on the OG produced by PNL (methyl-esterified) and the non-methyl-esterified OG degraded by PG? Is there an antagonism between their effects on the plant's immune response? This explanation in the supplementary figure would be useful to facilitate understanding.

Sentence on line 92: Can we talk about escape strategies in terms of plant immunity when *Botrytis cinerea* is also known to stimulate it and promote the cell death phenomenon necessary for necrosis?

Line 94-96: The sentence seems speculative to me.

Results:

Line 102: What do you mean by turnover?

Figure 1A: Although mentioned in the Materials and methods section, we would like to clarify the context here: this is a *Botrytis* culture in an Erlen shaker with detached leaves in minimal liquid medium. Can the presence of inorganic nitrogen and a simple sugar in the Gamborg medium (nitrate/ammonium and fructose) affect the expression results of the PNL genes of the fungus? Would a medium with no source of nitrogen and carbon other than organic nitrogen and polysaccharides from detached leaves have been possible? Would a fungus with a source of simple sugars and available inorganic nitrogen see these virulence genes induced even in the presence of detached leaves? I don't have the answer, but it intrigues me, and I wonder if the answer you're looking for wouldn't have been amplified without these extra resources. I'm not questioning the result obtained, which is significant, but the differences in OG production seem moderate (what %?). Do these differences seem biologically relevant to you? It's an explanation that the reader might expect. Perhaps we should say that although the effects seem moderate, it's the concordance of the results between the significant decrease in PG products and the significant increase in PNL products that arouses interest in the role of PNL in high nitrogen conditions.

For the gene expression measurements, the Bcin references are missing for all the genes analysed. I didn't find this information in the materials and methods, and it is necessary if people want to repeat the experiments.

Line 115-116: The underexpression of PNL genes seems to be contradicted by the overproduction of PNL products. Again, we lack the under-expression ratios.

The authors suggest that nitrogen nutrition has a greater effect on the structure of plant pectin than the fungal pectin degradation machinery. But couldn't we also imagine a post-transcriptional or post-translational up-regulation of PNLs?

I would simplify Figure 1 by removing Figure 1b, which I would move as an additional figure. I would keep in the main Figure 1 only the striking phenomena of inversions of PG and PNL products and significant expression variations in the plant genes. I would state in the text that the variations in PG and PNL products cannot be explained by transcriptional regulation of the PG and PNL nitrate/ammonium genes, and I would send the figure as a supplement.

Line 150: The argument is relevant, if the expression data is additionally moved, change the figure call here.

Line 159: Figure 2B (PCR) cannot be used to confirm whether or not there is ectopic integration because the cassette is bounded by regions of homologous recombination with no additional sequence on the edges that could be used for PCR. In the strategy used here, it is therefore the Southern blot (Figure 2C) that is used to confirm this. It is unfortunate that the Southern blot cannot be used to check the purity of the mutants. A different choice of probe would have been possible, even if the homologous recombination regions did not exceed 60pb (e.g. regions overlapping the homologous region). Indeed, it is possible that a thin band corresponding to the WT is not seen in the PCR but in the Southern...

I suggest moving Figures 2B and 2C (PCR and Southern) to the supplementary figures and keeping only the transformation strategy for Figure 2A in the main figure.

It seems incongruous to show a first phenotyping of the mutants without functional complementation and then a second phenotyping with complementation. A simplification seems necessary. I propose to move Figure 4B to the supplementary figures (PCR on complemented strains). I propose to merge Figure 2A with Figures 4A and 4D to have in the same figure the transformation strategies for KO and complementation and validation by RTqPCR. In Figure 4A, be sure to show the *fenR* gene, which disappears after recombination. Did the complementation strains screened for hygromycin resistance show a return of sensitivity to fenhexamid? Has this been tested? This would ensure that the complemented strains are pure and that no mutant nuclei remain.

Regarding Figure 4C, it seems redundant for the WT and KO mutant in Figure 3B. I therefore recommend merging Figures 3B and 4C to simplify the presentation of these results.

Lines 177-179: This seems to me to be a striking phenomenon of the PNL1 mutant, because the delay in penetration observed on the onion epidermis could explain the reduction in lesions observed on *Arabidopsis* leaves. This result is also used in the discussion to explain the overall low expression of plant defense genes when early infected by the mutant. If all the technical and biological replicates have been carried out and are consistent, it seems appropriate to move figure S2B to the main figure in the pathogenesis tests. The supplementary figure S2A would remain as a supplementary. The addition of a main figure is not problematic if other figures have been moved as supplementary as I proposed.

Figure 5B: If pectinases are globally underexpressed in the *pnl1* mutant, then the indicated fold change of expression should be that of the mutant versus WT (except that in the figure it is indicated WT versus mutant). Are all fold changes significant? The results of the statistical test for each FC (p-value?) are missing. What do the 'plant gene targets' in the legend correspond to, as I thought they were all fungal pectinases?

Figure 5C: For the same reasons as above, can the results obtained with the complemented strains be merged with the results for all the mutants in Figure 5A?

Is there a duplication of the WT results in Figure 1A and Figure 5A?

Figure 6: I think it would be more relevant if this figure appeared immediately after the pathogenicity tests and before the COS assay or pectinase expression.

Discussion:

Lines 319- 324: I don't understand this. It says that accumulation of PNL products correlates with induction of JA pathway repressors. Then in the next sentence PNL products induce the JA pathway (line 323: 'this pathway is induced')? This

seems contradictory to me. Can you clarify this?

In the Materials and Methods, for the construction of mutants :
Line 415 says that there is a figure in Supplement S3, but I haven't found it?

Version 1:

Reviewer comments:

Reviewer #1

(Remarks to the Author)

My concerns, major and minor, have been properly addressed by the authors. The precision is convincing. Additionally, all points raised by the reviewers improved the quality of the manuscript.

Reviewer #2

(Remarks to the Author)

The revised manuscript has been well considered and corrected the points raised in the previous review. However, there are some minor corrections that need to be made as indicated below.

Line 363-367

In Figure 5B, the amount of PNL product is not different between WT and Bcpl1 mutants under Low N conditions, so it is not clear whether the complementation of BcPNL1 restores the PNL product. Please improve the description.

Line 397-420.

The explanation of the results of the expression analysis is unclear and needs improvement.

In particular:

Line 401-402

At 2 dpi, the amounts of transcripts for AtPDF1.2a, AtPR4 and AtPAD3 increased compared to what? (Mock, WT, or 1 dpi?)

Line 405-406

What were the reduced transcript levels of AtJOX3, AtJAZ1, 3, and 4 compared to?

Line 412

B -> β

Line 412-420

Since the expression of AtPR2 and WRKY70 is relatively high in mock, comparisons with mock should be considered. For example, the expression of WRKY70 is described as increased in the Bcpl1 mutant at 1 dpi, but compared to Mock, it is unchanged or decreased. In other words, we can assume that a decrease in WRKY70 expression due to WT inoculation was weakened by inoculation with the Bcpl1 mutant. In addition, AtPR2 expression is altered in Mock, which may be an effect of using detached leaves. This needs to be considered as well.

Other points

1. line 169 "... decreased by 1.5 at high N ..."

Please check if the description is correct.

2. line 193-194

I think "(Figure 1B)" should be inserted after "respectively".

3. line 238 "CaCl₂"

The "2" should be subscripted.

4. line 296

...than that of BcPG1 (Figure 3H), ...

5. line 332

BCPNL1 -> BcPNL1

6. Materials and methods

Please follow the submission rules and state the units correctly throughout.

liters, L, l, $\mu\text{g/ml}$, $\mu\text{g ml}^{-1}$ etc.

Line 570 etc. 1×10^8 conida

Line 573 28°C

Line 858 1.5 kb, 0.5 kb

Line 643 3×10^5

7. References

Please check the submission rules throughout.

Line 681 7(8), e1002230

Line 697 12, 2166
Line 701 5, 435
Line 713 7, 1709
Line 716 9(11), 923
Line 730 17, 19
Line 786 PLoS Pathogens 16(8), e1008326
Line 790 10(2), 444
Line 827 11, 613259
Line 845 12, 782773
Line 853 Molecular Plant-Microbe Interactions, 35(10), 881–886
Line 895 BMC Research Notes, 3, 208
Line 899 8, e44279

Reviewer #3

(Remarks to the Author)

As most of my comments and requests for changes have been taken into account, I am in favour of publishing this revised version.

Version 2:

Reviewer comments:

Reviewer #2

(Remarks to the Author)

I confirm that the authors have adequately addressed all of the points I raised in the previous review in the revised manuscript. Nothing further to point out.

Editor: In particular, please note that the following revisions would be necessary for us to contact our referees again:

-Please add time-course experiments investigating the dynamics of the *B. cinerea* infection process over time (to address concerns raised by Reviewers #1 and #2). We would strongly recommend performing these new experiments in whole *Arabidopsis* plants (see the next point).

-Please thoroughly address concerns raised by the reviewers (particularly R1) about the use of detached leaves for experiments and how results obtained using this system may not reflect what happens in whole plants. If it is not possible to use whole plants for the assays, please explain why this is not possible and thoroughly discuss in the manuscript the caveats/limitations of using detached leaves (e.g., since detached leaves show dramatic changes in hormone accumulation and signaling, the results might not reflect what happens in leaves still attached to plants) and how choice of liquid medium might affect these results (as Reviewer #3 mentioned). Also along these lines, from an editorial perspective, we ask that you avoid using the phrase "in planta" for any experiments not performed using whole plants.

-Please address Reviewer #2's concern that the involvement of BcPNL1 in nitrogen-induced sensitivity may be negligible at best by either presenting additional data supporting this hypothesis or by toning down both your title and abstract to more accurately reflect the conclusions that can be drawn from the presented data.

-While we would welcome the addition of experiments investigating in more detail how the pectin structure of the host is altered by *B. cinerea* infection, as was requested by Reviewer #2, we will not require you to add these data to the revised manuscript.

First, we would like to thank the reviewers for their thorough and insightful review of our work, as well as for the time and effort they have invested in this process. Their feedback and constructive comments have greatly contributed to enhancing the clarity and quality of the manuscript. We have carefully addressed each of the reviewers' concerns point by point.

Referee:

Reviewer #1 plant defense, metabolites (Remarks to the Author):

General remarks and main concerns:

In this manuscript, Davière et al. present the impact of the *Botrytis cinerea* pectin lyase BcPNL1 in modulating *Arabidopsis thaliana* susceptibility under 2 nitrogen nutrition conditions. The manuscript is clear and the presented results are relevant. But my main concern is about the notion of response to infection which is not always observed at the same time after infection, it raises questions. Ideally it would be interesting to observe the dynamics of the infection with time-course analyses. But otherwise, the authors should at least justify why some analyses are done 1 day after infection, others 2 days after and also 15 or 18 hours after infection for LC-MS analyses.

We've added the following justification line 190 : "Since it has been shown that the majority of changes in plant gene expression occurs by 24 hpi when the pathogen has penetrated the leaf epidermis, and that, naturally, OG production precedes transcriptomic activation (Windram et al., 2012), we chose to delay the transcriptomic analysis slightly compared to the OG analysis and performed it at 1 dpi." and

added a second time point for both the transcriptomic analysis and the lesion area measurement (Figures 3, 5 and 6).

Another important point for me is that the study of the response is done on detached and incubated leaves. How can this biological model be consistent with the response of the leaf attached to the plant, particularly when looking at hormonal signalling?

Indeed, it has been shown that the entire infection process on living detached leaves was slower than disease development on attached leaves, resulting in the formation of a delayed and smaller lesion on detached leaves (Eizner et al., 2017). However, as described by the same authors, the differences in disease parameters between wild-type strain and strains that display reduced ability to cope with plant defence are increased on detached leaves, allowing us to have a better “visual” overview of the impact of *B. cinerea* gene inactivation on plant defense activation. Furthermore, although mass spectrometry is a very sensitive technique, the amount of biological infected material needed to detect OG remains consequent. That’s why we opted for liquid infection that allows to infect the entire leaves, releasing far more OGs than drop infection assays. This has been added lines 146-151.

Minor concerns

L 91 “GalA4MeAc-H2O” is an abbreviation, please give the full name **Done**.

L 124 “Since SA and JA defence signalling pathways are mutually antagonistic, ...”: add a reference. **Done**.

L 178 Could you evaluate the number of spores to compare the quality of the mutant versus the WT? **These data are already depicted Fig. S2A.**

L 204 Fig 3C What is the BcCUTA gene? **This has been added in the Fig. 3 legends.**

L 224 and Fig 5AC. The authors claim that “pectin degradation products in the presence of the Δ BcPnl1 mutants ... revealed a drastically reduced production of PNL derived OGs compared to the WT strain” but the data presented are expressed in peak area. It is best to use an absolute value when comparing a quantity, as each PG and NLP product standard may have different profiles.

We are not certain that we fully understand the reviewer's concern. The peak area (as opposed to relative peak area) is proportional to the absolute amount of each component, even though, in the absence of standards, these peak areas cannot be converted into molar equivalents. Therefore, while we cannot claim that there is a higher or lower amount of PG product compared to PNL product within the same sample, comparing the peak area of the same OG between two different samples is equivalent to comparing absolute values

Fig5B: At the top, what do the numbers mean?

The figure (now Fig. 5C) was modified to make clearer that these numbers are the gene numbers.

Reviewer #2 plant immunity, plant-pathogen (Remarks to the Author):

In this manuscript, the involvement of the pathogen pectin lyase (BcPnl1) in the nitrogen-induced susceptibility of the interaction between Arabidopsis and Botrytis cinerea, previously reported by the

authors, is analyzed using a pathogen mutant ($\Delta Bc\text{pnl}1$). The authors also analyzed changes in host pectin metabolism during *B. cinerea* infection and suggest that PNL-derived oligogalacturonides, whose accumulation increases under high nitrogen conditions, reduce host resistance levels via induction of expression of repressors of jasmonic acid signaling. They then propose that this may be involved in nitrogen-induced susceptibility. The experimental results are clear and suggestive to a certain degree. However, there are some points that must be carefully considered in order to support the conclusions proposed in this paper.

Major points

1. The main data in this paper are considered to be the results of the analysis using $\Delta Bc\text{pnl}1$. Since $\Delta Bc\text{pnl}1$ reduces virulence, there is no doubt that BcPNL1 has an important role as a pathogenesis factor in *B. cinerea*. However, as shown in Figure S2, due to the slow germination of spores in $\Delta Bc\text{pnl}1$, the subsequent infection process is delayed compared to WT. It is highly possible that this timing discrepancy will also affect subsequent host responses. Unfortunately, each of the experiments shown was performed at a single time point and does not account for the staggered timing of infection. To solve this problem, experiments using $\Delta Bc\text{pnl}1$ should show detailed changes in the infection process over time, and then changes in gene expression should be shown in time course.

That's an excellent point. To address this concern, we added lesion area results at 72 hpi results Fig3 and gene expression results at 2 dpi Fig5 and 6. The corresponding text has been added.

2. The authors concluded that high nitrogen affected the host pectin structure, since the accumulation of PNL products increased even though the expression of PNL in *B. cinerea* was not increased under high nitrogen conditions (lines 111-116, 329 -332 lines). I agree with this. However, it is not clear exactly how the pectin structure of the host was altered. Additional experiments should be conducted or the experimental data and other findings should explain what the possibilities are.

We added data showing that PNL products are exclusively detected when *B. cinerea* spores are incubated with acetyl- and methyl-esterified pectins while no PNL products are detected when spores are incubated with methylesterified but non-acetylerified pectins suggesting that the nitrogen could impact acetylerification state of the pectins. We added these results Fig. S3 and line 176-185.

3. The authors state in lines 184-187 and 364-366 that BcPNL1 alone cannot explain nitrogen-induced susceptibility. The authors claim that BcPNL1 is involved in nitrogen-induced sensitivity because JAZ gene expression is not significantly different between high and low nitrogen conditions in the mutant in Figure 7B, but the data are not shown in detail enough to make an objective determination.

The involvement of BcPNL1 in nitrogen-induced sensitivity may be negligible, if any. If so, the title and abstract may be misleading to the reader.

By adding 48hpi data as requested (Fig. 5 and Fig. 6), this claim is more supported. Indeed, the difference at 48hpi of *JAZ1*, *JAZ3*, *JAZ4* and *JOX3* expression between high N and low N infected plants becomes more pronounced than at 24hpi and we did not observed such a difference in High N and low N plants infected by $\Delta Bc\text{pnl}1$ mutant.

Minor points.

1. Why was the analysis performed at 18hpi in Figure 5C?

To avoid being below the detection limit with the mutant strains, we decided to perform the experiment a bit latter, where more OGs are accumulated in the liquid medium.

2. Line 239-241. I don't know why the low PG product of Δ Bcpl1.1c2 confirms that PNL is essential prior to PG activity.

We added a figure in supplemental data (Fig S1) to describe the pectin degradation process.

Reviewer #3 Botrytis cinerea (Remarks to the Author):

Few studies in plant pathology have considered all components of the disease triangle, i.e. pathogen virulence, host response and environmental effects. However, abiotic stresses certainly have a major influence on biotic stresses, which is still largely unknown. By investigating the effect of plant nitrogen fertilisation on disease development by the Botrytis fungus, this study shows that the degradation of plant pectin by the fungus is regulated as a function of plant nitrogen nutrition. As well as identifying a new virulence factor in this fungus, this study has the wider merit of drawing our attention to the need to consider nitrogen fertilisation and, more generally, the impact of the environment on the virulence of plant pathogenic fungi. Because of its innovative aspect, I am rather in favour of the publication of this article, but a revision seems to me to be necessary.

Abstract :

The abstract needs to be rewritten and would benefit from being more factual.

There is a lack of precision in "revealed a noteworthy impact of nitrogen availability ... on PNL1". Is this a positive or negative effect? I wouldn't mention the mutant in the same sentence as it confuses the information. The nature of the effect is then explained, but this introduces redundancy that is not expected in a summary. I don't think we can say that the enzymatic activity of PNL1 is affected. This implies that the effect is direct, but there is no specific PNL1 enzymatic assay to verify this. This can be observed indirectly by deleting the gene that encodes this activity and measuring OG degradation products in the mutant. These degradation products may originate from PNL1 as well as from other PNLs and there may be epistatic relationships between the genes in this family.

We adjusted the abstract accordingly.

Introduction:

I did not find the review by Cheung et al. 2020 in the list of references. Please check again all references.

It has been added line 717.

It says that Botrytis has a high degree of functional redundancy and compensation phenomena, but these characteristics do not seem to be specific to this fungus and do not apply to all genes. I would rather explain that among the virulence factors, some are encoded by multi-gene families such as CAZY,

which are subject to functional redundancy and compensation phenomena that make them difficult to study in fungi.

It has been modified lines 39-42.

I wonder if there isn't a misunderstanding about the role of chitin synthase activities such as chs3a. It is said that they activate plant defence, but according to the title of the article cited, it would be the absence of chs3a in a deletion mutant that would induce the response.

Line 46, "Activating" has been replaced by "influencing".

The paragraph between lines 52-70 explains the enzymatic arsenal of the fungus dedicated to the degradation of plant pectin. It's complex and I think a scheme as a supplementary figure would be helpful, especially to understand the differences between PNL and PL activities. It is not specified how many genes are listed for each type of pectinolytic activity in *Botrytis*.

Lines 60-86. A scheme has been added (Fig. S1) and the paragraph has been enriched and rephrased.

The paragraph between lines 71 and 91 explains what is known about the plant's perception of OGs and its immune response. Could we have some clarification on the OG produced by PNL (methylesterified) and the non-methylesterified OG degraded by PG? Is there an antagonism between their effects on the plant's immune response? This explanation in the supplementary figure would be useful to facilitate understanding.

Lines 108-110, a sentence has been added and hopefully, the scheme Fig. S1 should facilitate understanding.

Sentence on line 92: Can we talk about escape strategies in terms of plant immunity when *Botrytis cinerea* is also known to stimulate it and promote the cell death phenomenon necessary for necrosis?

We agree. We modified this sentence by "To gain deeper insight into the strategies employed by *B. cinerea* to successfully infect plants" (line 111).

Line 94-96: The sentence seems speculative to me.

Line 113, we replaced "Our research provides evidence that nitrogen fertilization impacts *A. thaliana* pectin sensitivity to PNLs and that BcPNL1 contributes to the factors linking disease susceptibility and nitrogen fertilization" by "Our research provides compelling evidence that nitrogen fertilization influences pectin degradation in *A. thaliana* detached leaves infected by *B. cinerea*."

Results:

Line 102: What do you mean by turnover?

Line 142, a definition of turnover has been added (OG production and subsequent degradation into monomers).

Figure 1A: Although mentioned in the Materials and methods section, we would like to clarify the context here: this is a Botrytis culture in an Erlen shaker with detached leaves in minimal liquid medium. Can the presence of inorganic nitrogen and a simple sugar in the Gamborg medium (nitrate/ammonium and fructose) affect the expression results of the PNL genes of the fungus? Would a medium with no source of nitrogen and carbon other than organic nitrogen and polysaccharides from detached leaves have been possible? Would a fungus with a source of simple sugars and available inorganic nitrogen see these virulence genes induced even in the presence of detached leaves? I don't have the answer, but it intrigues me, and I wonder if the answer you're looking for wouldn't have been amplified without these extra resources.

We agree. We have added a sentence in lines 153-156 to make the scope of the study clearer. "It is worth noting that the minimal liquid medium used contains 25 mM NO₃⁻. Consequently, the phenomena we observed correspond to the impact of N nutrition on the plant, notably its cell wall structure, that in turn affects the infection process rather than the impact of N availability on fungal virulence."

I'm not questioning the result obtained, which is significant, but the differences in OG production seem moderate (what %?). Do these differences seem biologically relevant to you? It's an explanation that the reader might expect. Perhaps we should say that although the effects seem moderate, it's the concordance of the results between the significant decrease in PG products and the significant increase in PNL products that arouses interest in the role of PNL in high nitrogen conditions.

We acknowledge that the initial presentation of the results may make the overall difference appear moderate. However, a closer examination reveals that the effect for individual DPs is quite substantial. To emphasize this, we have added the percentage differences for the differentially accumulated OGs in the text and updated the graph Fig. 1 to make the results more striking.

For the gene expression measurements, the Bcin references are missing for all the genes analysed. I didn't find this information in the materials and methods, and it is necessary if people want to repeat the experiments.

This has been added in the figure S2 and table S2.

Line 115-116: The underexpression of PNL genes seems to be contradicted by the overproduction of PNL products. Again, we lack the under-expression ratios.

We have added the PNL expression in liquid medium (Fig. S2B) which corroborates the fact that the availability of nitrogen might impact PNL expression and the ratio has been added in the text line 171

The authors suggest that nitrogen nutrition has a greater effect on the structure of plant pectin than the fungal pectin degradation machinery. But couldn't we also imagine a post-transcriptional or post-translational up-regulation of PNLs?

Indeed, we cannot rule out the possibility of post-transcriptional or post-translational regulation of PNLs. This has been added lines 499-500.

I would simplify Figure 1 by removing Figure 1b, which I would move as an additional figure. I would keep in the main Figure 1 only the striking phenomena of inversions of PG and PNL products and significant expression variations in the plant genes. I would state in the text that the variations in PG and PNL products cannot be explained by transcriptional regulation of the PG and PNL nitrate/ammonium genes, and I would send the figure as a supplement.

This has been done in the text and the figures.

Line 150: The argument is relevant, if the expression data is additionally moved, change the figure call here.

Line 159: Figure 2B (PCR) cannot be used to confirm whether or not there is ectopic integration because the cassette is bounded by regions of homologous recombination with no additional sequence on the edges that could be used for PCR. In the strategy used here, it is therefore the Southern blot (Figure 2C) that is used to confirm this.

We agree, this confusion came from an error in the figure referenced in the text. This has been added Fig. S4.

It is unfortunate that the Southern blot cannot be used to check the purity of the mutants. A different choice of probe would have been possible, even if the homologous recombination regions did not exceed 60pb (e.g. regions overlapping the homologous region). Indeed, it is possible that a thin band corresponding to the WT is not seen in the PCR but in the Southern...

We agree but the probability of such a phenomenon is very low and the CRISPR-Cas9 system in *Botrytis cinerea* was shown to be very efficient.

I suggest moving Figures 2B and 2C (PCR and Southern) to the supplementary figures and keeping only the transformation strategy for Figure 2A in the main figure.

This has been done as requested.

It seems incongruous to show a first phenotyping of the mutants without functional complementation and then a second phenotyping with complementation. A simplification seems necessary. I propose to move Figure 4B to the supplementary figures (PCR on complemented strains). I propose to merge Figure 2A with Figures 4A and 4D to have in the same figure the transformation strategies for KO and complementation and validation by RTqPCR.

Figures were modified as requested.

In Figure 4A, be sure to show the *fenR* gene, which disappears after recombination. Did the complementation strains screened for hygromycin resistance show a return of sensitivity to fenhexamid? Has this been tested? This would ensure that the complemented strains are pure and that no mutant nuclei remain.

It seems that there is a misunderstanding regarding the recombination strategies used to produce the complemented strains. It would have been possible to use the *fenR* locus for recombination and then use the loss of fenhexamide resistance as a sign of nuclei purity but we preferred to follow a recently published

paper where an intergenic region in chromosome 3 was used successfully to insert an overexpression cassette (Jeblick *et al.*, 2023). We agree on the lack of one PCR result showing the disappearance of the deleted WT sequence after recombination and we therefore added the results in figure S4C.

Regarding Figure 4C, it seems redundant for the WT and KO mutant in Figure 3B. I therefore recommend merging Figures 3B and 4C to simplify the presentation of these results.

They cannot be merged because all mutants and complemented lines have not been analyzed at the same time.

Lines 177-179: This seems to me to be a striking phenomenon of the PNL1 mutant, because the delay in penetration observed on the onion epidermis could explain the reduction in lesions observed on Arabidopsis leaves. This result is also used in the discussion to explain the overall low expression of plant defense genes when early infected by the mutant. If all the technical and biological replicates have been carried out and are consistent, it seems appropriate to move figure S2B to the main figure in the pathogenesis tests.

We have moved the Figure S2B to Figure 4A and Fig4B. We have also added a comparison of *Bcpn1* pathogenicity with *Bcpg1* and *Bcpme1/2* on Arabidopsis (lines 296-305)

The supplementary figure S2A would remain as a supplementary. The addition of a main figure is not problematic if other figures have been moved as supplementary as I proposed.

Figure 5B: If pectinases are globally underexpressed in the *pnl1* mutant, then the indicated fold change of expression should be that of the mutant versus WT (except that in the figure it is indicated WT versus mutant).

Thank you for pointing that out. The error has been fixed.

Are all fold changes significant? The results of the statistical test for each FC (p-value?) are missing.

In the heat map representing the fold changes red and green boxes are significant whereas grey boxes are not. The p-value threshold is indicated in the legend.

What do the 'plant gene targets' in the legend correspond to, as I thought they were all fungal pectinases?

Text error has been fixed.

Figure 5C: For the same reasons as above, can the results obtained with the complemented strains be merged with the results for all the mutants in Figure 5A?

Is there a duplication of the WT results in Figure 1A and Figure 5A?

The WT results in Figures 1A and 5A come from two independent experiments. While they show similar trends, they were conducted separately to ensure the reliability and reproducibility of the findings.

Figure 6: I think it would be more relevant if this figure appeared immediately after the pathogenicity tests and before the COS assay or pectinase expression.

As requested, the Figure 6 has been moved to Figure 4.

Discussion:

Lines 319- 324: I don't understand this. It says that accumulation of PNL products correlates with induction of JA pathway repressors. Then in the next sentence PNL products induce the JA pathway (line 323: 'this pathway is induced')? This seems contradictory to me. Can you clarify this?

The sentence (line 489) has been modified in order to make it clearer.

In the Materials and Methods, for the construction of mutants :

Line 415 says that there is a figure in Supplement S3, but I haven't found it?

It was a mistake. Now Figure 2 shows the knockout strategies and Figure S4 the diagnostic PCRs.

Reviewers' comments:

Reviewer #2 (Remarks to the Author):

The revised manuscript has been well considered and corrected the points raised in the previous review. However, there are some minor corrections that need to be made as indicated below.

We sincerely appreciate the reviewer's thorough and exhaustive evaluation of our manuscript. Below, we provide our detailed responses to the comments.

Line 363-367

In Figure 5B, the amount of PNL product is not different between WT and Bcpl1 mutants under Low N conditions, so it is not clear whether the complementation of BcPNL1 restores the PNL product. Please improve the description.

We've improved the description by modifying the text starting from line 348 as follows: "No clear conclusion could be formulated regarding PNL products for which only small DP were detected suggesting that, at this time point, most of the PNL products were already degraded. However, we observed with the *ΔBcpl1.1c* strains the restoration of PG products accumulation for DP3 and DP4 in both N conditions and DP5 in low N that confirms that PNL activity is essential prior to PG and PME activity."

Line 397-420.

The explanation of the results of the expression analysis is unclear and needs improvement.

In particular:

Line 401-402

At 2 dpi, the amounts of transcripts for AtPDF1.2a, AtPR4 and AtPAD3 increased compared to what? (Mock, WT, or 1 dpi?)

From lines 385 to 388, we've modified the text as follows: "However, at 2 dpi, transcript levels in detached leaves infected with the *ΔBcpl1* mutants increased in all cases compared to 1 dpi, particularly under high N conditions. In this scenario, *AtPDF1.2a* expression was notably more elevated upon infection with the *ΔBcpl1* mutants than with the WT."

Line 405-406

What were the reduced transcript levels of AtJOX3, AtJAZ1, 3, and 4 compared to?

We've added "compared to the WT" line 392.

Line 412-420

Since the expression of AtPR2 and WRKY70 is relatively high in mock, comparisons with mock should be considered. For example, the expression of WRKY70 is described as increased in the Bcpl1 mutant at 1 dpi, but compared to Mock, it is unchanged or decreased. In other words, we can assume that a decrease in WRKY70 expression due to WT inoculation was weakened by inoculation with the Bcpl1 mutant. In addition, AtPR2 expression is altered in Mock, which may be an effect of using detached leaves. This needs to be considered as well.

From lines 398 to 411, we've modified the text as follows: "The expression profile of the *AtPR2* gene, coding for a β -1,3-glucanase with antifungal properties also involved in callose deposition and SA-

dependent defense responses^{43,44}, indicates that the presence of *ΔBcpn11* mutants on detached leaves triggered a plant response linked to SA at 1 dpi in high N which is absent in mock or upon inoculation with the WT. At 2dpi, *AtPR2* expression was induced in the high N mock compared to 1 dpi, which may be an effect of using detached leaves. By contrast, *AtPR2* expression was downregulated upon infection compared to the mock in a similar way with the WT and the mutants. (Figure 6). Finally, the expression of the *AtWRY70* gene, which is involved in SA and JA/ethylene signaling pathways^{36,37}, was repressed upon infection compared to the mock at 1 dpi suggesting the presence of JA. This gene repression was notably higher with the WT compared to the *ΔBcpn11* mutants in both N regimes. At 2 dpi, *AtWRY70* expression was still repressed by the infection but interestingly the repression was higher with *ΔBcpn11* mutants compared to the WT under high N conditions which might reflect a delay in *AtWRKY70* repression with *ΔBcpn11* mutants. This indicates that SA and JA/ET signaling is modified during the kinetics of *B. cinerea* infection to tend to produce JA and this process is influenced by N levels and the absence of *BcPNL1*.”

Other points

1. line 169 “... decreased by 1.5 at high N ...”
Please check if the description is correct.

The description is correct, the value for BcPNL1 expression in low N is around 1.2 and 0.8 in high N.

All the following modifications have also been made. Additionally, we reduced the abstract and replaced "authors et al." with numerical citations to comply with the submission guidelines. The line numbers have been updated.

2. Line 181
I think “(Figure 1B)” should be inserted after “respectively”.

3. Line 226
The “2” should be subscripted.

4. Line 283
...than that of BcPG1 (Figure 3H), ...

5. Line 317
BCPNL1 -> BcPNL1

6. Line 398
B -> β

6. Materials and methods
Please follow the submission rules and state the units correctly throughout.
liters, L, l, μg/ml, μg ml⁻¹ etc.
Line 570 etc. 1×10⁸ conida
Line 573 28°C
Line 858 1.5 kb, 0.5 kb
Line 643 3×10⁵

Done

7. References

Please check the submission rules throughout.

Line 681 7(8), e1002230

Line 697 12, 2166

Line 701 5, 435

Line 713 7, 1709

Line 716 9(11), 923

Line 730 17, 19

Line 786 PLoS Pathogens 16(8), e1008326

Line 790 10(2), 444

Line 827 11, 613259

Line 845 12, 782773

Line 853 Molecular Plant-Microbe Interactions, 35(10), 881–886

Line 895 BMC Research Notes, 3, 208

Line 899 8, e44279